# Accelerating Optimization and Machine Learning through Decentralization

## Abstract

Decentralized optimization enables multiple devices to learn a global machine learning model while each individual device only has access to its local dataset. By avoiding the need for training data to leave individual users' devices, it enhances privacy and scalability compared to conventional centralized learning where all Data has to be aggregated to a central server. However, decentralized optimization has traditionally been viewed as a necessary compromise, used only when centralized processing is impractical due to communication constraints or data privacy concerns. In this study, we show that decentralization can paradoxically accelerate convergence, outperforming centralized methods in the number of iterations needed to reach optimal solutions. Through examples in logistic regression and neural network training, we demonstrate that distributing data and computation across multiple agents can lead to faster learning than centralized approaches—even when each iteration is assumed to take the same amount of time, whether performed centrally on the full dataset or decentrally on local subsets. This finding challenges longstanding assumptions and reveals decentralization as a strategic advantage, offering new opportunities for more efficient optimization and machine learning.

## 1 Introduction

We focus on the fundamental task of mathematical optimization—the process of iteratively refining a solution to minimize a loss function with respect to a given metric. In typical application domains such as machine learning, engineering, and operations research, the loss is usually defined on a large number of samples or measurements collected from multiple locations (LeCun et al., 2015). For example, the development of machine learning based disease classifiers needs a large amount of training data to ensure accuracy and minimize bias (Savage, 2017; Warnat-Herresthal et al., 2021; Peterson et al., 2021; Wang et al., 2023). In sensor network based environment sensing and climate monitoring, a sufficient amount of measurement data is needed to suppress the noise inherent in low-cost sensors (Iyer et al., 2022; Beucler et al., 2024; Bracco et al., 2025). Although a natural approach to optimization or learning on such data is to aggregate everything on a central server and train a model directly, limitations in computational and storage capacity on individual devices have led to growing interest in decentralized learning methods. These methods are uniquely suited to handle the rapid growth in data volume and model complexity in modern training tasks (Jiang et al., 2017; Koloskova et al., 2019). Moreover, training data may sometimes involve sensitive information, such as medical records, and legislation may prohibit aggregating data from distributed sources like hospitals (Warnat-Herresthal et al., 2021). In such cases, decentralized learning becomes the only viable option as it allows the training data to stay on individual learning devices while enabling them to learn a common global optimal solution.

In the past decade, substantial progress has been made towards developing efficient decentralized machine learning algorithms, both in the fully decentralized server-free case (which is also called consensus optimization (Nedic et al., 2010; Shi et al., 2015; Koloskova et al., 2019)) and the parameter-server assisted case (which is also called federated optimization (McMahan et al., 2017; Kairouz et al., 2021) or cloud-based optimization (Nedić & Liu, 2018)). However, it is generally believed that such decentralized learning cannot match the performance of its centralized counterpart, where a powerful server performs computation directly

on aggregated data. (It is important to note that the commonly cited "linear speedup" in distributed optimization with an increasing number of computing devices, as shown by Lian et al. (2017) and Yu et al. (2019), assumes that each device has fixed computing power, so that adding more devices effectively increases total computational capacity. This assumption is orthogonal to our setting, where the central server has computing power equivalent to the combined capacity of all distributed devices.) In this paper, we challenge this widely held belief and demonstrate—both theoretically and empirically—that decentralization can accelerate optimization by reducing the number of iterations required compared to centralized methods.

## 2 Problem statement and context

### 2.1 Preliminaries

**Definition 1** (Strongly convex). *A function $f : \mathbb{R}^d \to \mathbb{R}$ is strongly convex if there exists a constant $\mu > 0$ such that for all $x, y \in \mathbb{R}^d$, the following inequality holds: $f(y) \geq f(x) + \nabla f(x)^\top (y - x) + \frac{\mu}{2}\|y - x\|^2$.*

**Definition 2** (Lipschitz smoothness). *A function $f : \mathbb{R}^d \to \mathbb{R}$ is Lipschitz smooth with constant $L > 0$ if for all $x, y \in \mathbb{R}^d$, the following inequality holds: $\|\nabla f(x) - \nabla f(y)\| \leq L\|x - y\|$.*

### 2.2 Optimization problem

Let $\mathcal{D} = \{\xi_1, \xi_2, \cdots, \xi_{|\mathcal{D}|}\}$ represent a set of data samples with $\xi_i$ denoting the $i$th data sample and $|\mathcal{D}|$ denoting the size of the dataset. Then, most optimization or machine learning tasks on the dataset $\mathcal{D}$ can be mathematically formulated as the following problem:

$$\min_{x \in \mathcal{X}} f(x) := \frac{1}{|\mathcal{D}|} \sum_{i=1}^{|\mathcal{D}|} \ell(x, \xi_i), \tag{1}$$

where $\mathcal{X}$ denotes the set of all models/solutions that are of interest, $\ell$ denotes the loss function and $\ell(x, \xi_i)$ represents the loss of a model $x$ evaluated on data sample $\xi_i$. For example, in the popular empirical risk minimization problem (Vapnik, 2006), $\xi_i$ is of the form of $(a_i, b_i)$ with $a_i$ denoting the input (feature) and $b_i$ denoting the output (label). A model $x$ predicts an output $x(a_i)$ and the precision of this prediction, that is, the distance between the predicted output $x(a_i)$ and the actual output $b_i$, is measured by the loss $\ell(x, \xi_i) = \ell(x, (a_i, b_i))$. Commonly used models $x$ include linear/logistic regression models and neural networks. And commonly used examples of the loss function include cross-entropy functions, Kullback–Leibler (KL) divergence, regression functions, and many others (Shalev-Shwartz & Ben-David, 2014).

### 2.3 Optimization algorithms

To solve the mathematical optimization problem in (1), many algorithms have been proposed, including gradient descent methods (e.g., Bertsekas & Tsitsiklis (2000); Nesterov (2013)), distributed alternating direction method of multipliers (e.g., Boyd et al. (2011)), Newton's methods (e.g., Dennis & Moré (1977)), and many others. All of these methods are iterative, i.e., they start from some initial guess of the optimal solution and then repeatedly refine the guess using sampled data under a given rule. We focus on gradient methods since they (as well as their stochastic variants) are among the most widely used due to their efficiency in both computational complexity and storage requirements. In fact, gradient methods are also widely regarded as the current workhorse and cornerstone for the training of modern neural networks (Assran et al., 2019).

We focus on the standard gradient descent method, which takes the following form:

$$x^{k+1} = x^k - \alpha g^k, \tag{2}$$

where $x^k$ denotes the model value in the $k$th iteration, $\alpha$ is a positive scalar denoting the step size, and $g^k$ denotes the gradient evaluated at $x^k$. In machine learning applications, usually a stochastic estimate of the gradient (based on mini-batch sampling of available data) is employed. When the objective function is convex and $L$-smooth, i.e., there is a constant $L$ such that $\|g(x) - g(x')\| \leq L\|x - x'\|$ holds for all $x, x' \in \mathcal{X}$, it is well known that a stepsize $\frac{1}{L}$ gives guaranteed convergence to an optimal solution.

Recent years have also witnessed the development of numerous decentralized gradient methods that enable multiple devices to learn a common optimal solution while each device can only have access to a subset of the total dataset (see, e.g., Nedic & Ozdaglar (2009); Shi et al. (2015); Di Lorenzo & Scutari (2015); Nedic et al. (2017); Qu & Li (2017); Xu et al. (2017); Karimireddy et al. (2020)). In these studies, the objective function is usually assumed to be of a form $f(x) = \sum_{i=1}^{N} f_i(x)$ with $N$ denoting the number of participating devices and $f_i(x)$ denoting the loss of the subset of data only accessible to device $i$. There are two commonly studied decentralized gradient methods: server-assisted decentralized gradient methods (see, e.g., McMahan et al. (2017); Karimireddy et al. (2020); Xiang et al. (2024)) and server-free decentralized gradient methods (see, e.g., Nedic & Ozdaglar (2009); Allouah et al. (2024)). In the former method, a central server is responsible for updating the optimization variable, while each participating device $i$ is only tasked with computing gradients $g_i^k$ based on its local function $f_i$ and the variable $x$ received from the server. In this case, the update usually takes the form $x^{k+1} = x^k - \alpha \frac{\sum_{i=1}^{N} g_i^k}{N}$. In the latter, server-free case, each participating agent maintains a local copy of $x$, denoted $x_i$. Each device updates its local copy $x_i$ based on information received from its neighbors and its local gradient $g_i$. Assuming that device $j$ shares $w_{ij}(x_j^k - \alpha g_j^k)$ at the $k$th iteration to its neighbors with $w_{ij} \geq 0$ denoting the influence of device $j$ on device $i$, the update of every device $i$ takes the form $x_i^{k+1} = \sum_{j=1}^{N} w_{ij}(x_j^k - \alpha g_j^k)$. It is worth noting that when the coupling coefficient $w_{ij}$ is set to $\frac{1}{N}$ for all $1 \leq i, j \leq N$, the dynamics of the server-free decentralized optimization become equivalent to those of the server-assisted case when the initial values of all devices (i.e., $x_i^0$ for all $i$) are set as the same value.

It is worth noting that although those distributed versions of gradient methods allow (1) to be solved in a distributed manner with each participating device only having a subset of the total data, they generally require more iterations to reach a given accuracy level (or, at best, perform on par) compared to their centralized counterparts, in which a single device performs gradient descent on the full dataset.

---

**Algorithm 1**

---

1: **Initialization:** $x^0$, step size $\alpha_i = \frac{1}{L_i}$ for device $i$, switching threshold $\epsilon > 0$, switching state $\mathcal{S} = F$.
2: **for** $k = 1, \cdots, K$ **do**
3:    **if** $\| \sum_{i=1}^{N} \alpha_i g_i^k \| \leq \epsilon$ and $\mathcal{S} = F$ **then**
4:       Set $\mathcal{S} = T$ and reset $\alpha_i = \frac{1}{L}$ with $L = \frac{\sum_{i=1}^{N} L_i}{N}$;
5:    **end if**
6:    Each device receives $x^{k-1}$ from the server, computes $g_i^{k-1}$, and then sends it back to the server;
7:    The server updates
$$x^k = x^{k-1} - \frac{\sum_{i=1}^{N} \alpha_i g_i^{k-1}}{N};$$
8: **end for**

---

# 3 Decentralization approach and theoretical performance evaluation

## 3.1 Decentralization approach

When data are generated across different devices and locations, a natural approach to solving optimization problem (1) is to aggregate all data on a central server and then apply a centralized optimization method, such as the centralized gradient method. In fact, since data collected from different locations often exhibit heterogeneous distributions—for example, facial images captured by cameras tend to reflect the demographics of each location, and images of kangaroos are typically collected only from cameras in Australia or zoos (Hsieh et al., 2020)—this heterogeneity poses a significant challenge to decentralized learning algorithms (Hsieh et al., 2020). Consequently, it is widely believed that aggregating all datasets and performing learning centrally is a more desirable approach. However, we argue that when local devices possess computing capabilities, retaining data on the devices and solving problem in (1) via decentralized optimization can lead to faster convergence—especially in cases where the data distribution is skewed across devices. More specifically, we demonstrate that when data across different devices follow heterogeneous distributions, the

local geometric properties of these datasets can be leveraged to accelerate optimization by reducing the number of iterations needed for solving (1).

Our key idea for exploiting the local geometry of individual objective functions is to employ a server-assisted decentralized optimization algorithm, where each device uses a local step size tailored to the smoothness constant of its own dataset. Specifically, represent the dataset on device $i$ as $\mathcal{D}_i$. Then device $i$'s local loss function is given by $f_i(x) := \frac{1}{|\mathcal{D}_i|} \sum_{j=1}^{|\mathcal{D}_i|} \ell(x, \xi_j)$ and we can obtain its local smoothness constant $L_i$ and set its local step size as $\frac{1}{L_i}$. Note that the definition of the loss function $f(\cdot)$ in (1), which is the average over all data samples, implies $L = \frac{\sum_{i=1}^{N} |\mathcal{D}_i| L_i}{\sum_{i=1}^{N} |\mathcal{D}_i|}$ (which reduces to $L = \frac{\sum_{i=1}^{N} L_i}{N}$ when the sizes of datasets $|\mathcal{D}_i|$ are equal). Of course, with each device $i$ using a heterogeneous step size $\frac{1}{L_i}$, different devices' gradients are weighted unevenly in the update of $x$, which will result in a deviation from the original optimal solution (Wang & Nedić, 2023). To mitigate the resulting optimization errors, we propose switching from heterogeneous step sizes to a universal step size, $\frac{1}{L}$, once convergence plateaus under the heterogeneous step size regime. This strategy can ensure convergence to an exact optimal solution (as confirmed by our theoretical evaluation in Fig. 1). The detailed algorithm is summarized in Algorithm 1.

It is worth noting that even when the data are already centralized on a server, this approach can still be applied by intentionally partitioning the data in a non-uniform manner—for example, by dividing it according to data labels.

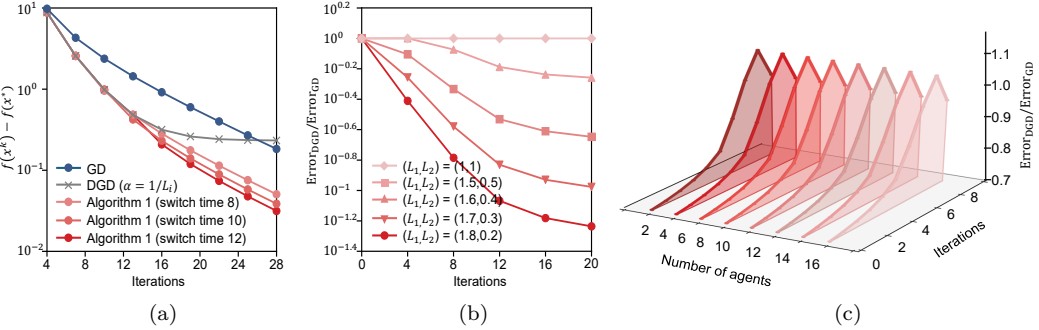

Figure 1: Performance comparison of Algorithm 1 with its centralized counterpart (GD) using PEP based theoretical analysis. (a) shows the acceleration of Algorithm 1 (results under three switch time instants are given, represented by three red curves with varying shades) over its centralized counterpart (termed GD and represented by the blue curve). The centralized approach uses a step size $\frac{1}{L}$ with $L = \frac{1}{N} \sum_{i=1}^{N} L_i$. The gray curve represents the performance of the conventional decentralized gradient descent (termed DGD) algorithm with local step sizes, i.e., agent $i$ using a step size $\frac{1}{L_i}$, which is subject to a steady-state error. In this subplot, we consider two agents with heterogeneous local smoothness constants $L_1 = \frac{1}{3}$ and $L_2 = 3$. (b) demonstrates that a greater heterogeneity between $L_1$ and $L_2$ leads to more acceleration via decentralization. The y-axis represents the ratio of optimization errors obtained by PEP under Algorithm 1 and GD. A ratio less than 1 indicates Algorithm 1 gaining acceleration over its centralized counterpart. (c) considers the more than two-agent setting with half of the agents having a smoothness constant of $\frac{1}{3}$, whereas the other half have a smoothness constant of 3. The strongly convex parameter is set to $\mu = 0.1$.

### 3.2 Theoretical evaluation

Recently, a computational framework known as the Performance Estimation Problem (PEP) (Drori & Teboulle, 2014; Taylor et al., 2017b) was introduced to precisely characterize the worst-case performance of optimization algorithms. In contrast to traditional asymptotic analyses, which often provide conservative or loose guarantees, PEP yields exact performance bounds for algorithms over specified function classes.

To describe the PEP framework, we first define the augmented states as $x^k = [x_1^k, x_2^k, \ldots, x_N^k] \in \mathbb{R}^{d \times N}$, where the superscript $k$ belongs to the index set $I_K = \{0, \ldots, K\}$. Similarly, we define the augmented local optimum and global optimum as $x^\star = [x_1^\star, x_2^\star, \ldots, x_N^\star]$ and $x^* = [x_1^*, x_2^*, \ldots, x_N^*]$, respectively

(where $x_1^* = x_2^* = \cdots = x_N^*$ will be enforced in the PEP formulation). To simplify notation, we extend the index set $I_K$ for augmented states to include these two optimal states, defining the augmented index set as $I_K^{\star,*} = \{0, \ldots, K, \star, *\}$.

In a similar manner, we define the augmented gradients and function values as

$$g^k = [g_1^k, \ g_2^k, \ \ldots, \ g_N^k] \in \mathbb{R}^{d \times N}, \quad f^k = [f_1^k, \ f_2^k, \ \ldots, \ f_N^k] \in \mathbb{R}^N, \quad \text{for } k \in I_K^{\star,*}.$$

We use $\mathcal{F}_{\mu,L}$ to denote the class of functions such that each $f \in \mathcal{F}_{\mu,L}$ is $\mu$-strongly convex and $L$-smooth.

Under these definitions and denoting $\mathbb{S}_+$ as the set of all symmetric positive semidefinite matrices, we have

$$P = [x^0, \ g^0, \ g^1, \ \ldots, \ g^K, \ g^\star, \ g^*, \ x^\star, \ x^*] \in \mathbb{R}^{d \times [(K+6)N]},$$
$$G = P^\top P \in \mathbb{S}_+^{((K+6)N) \times ((K+6)N)}, \ F = [f^0, \ f^1, \ \ldots, \ f^K, \ f^\star, \ f^*] \in \mathbb{R}^{1 \times (K+3)N},$$

which allows us to formulate the PEP as:

$$\max_{\substack{x^0, \ g^0, \ g^1, \ \ldots, \ g^K, \\ g^\star, \ g^*, x^\star, \ x^*}} f(\bar{x}^K) - f(x^*)$$

s.t. $\{x_i^k, \ g_i^k, \ f_i^k\}, \ k \in I_K^{\star,*}$ are interpolated for each local function class $\mathcal{F}_{\mu_i, L_i}$,

$\{x_i^k\}_{i \in [N], \ k \in I_K}, \ \bar{x}^K$ are generated recursively by algorithm 1 and $x_1^k = x_i^k, \forall i \in [N], k \in I_K$,

$\dfrac{1}{N}\sum_{i=1}^N g_i^* = \dfrac{1}{N}\sum_{i=1}^N \nabla f_i(x^*) = 0$, such that $x^*$ is the global optimum and $x_1^* = x_i^*, \forall i \in [N]$,

$g_i^\star = \nabla f_i(x_i^\star) = 0, \ \forall i \in [N]$, such that $x_i^\star$ is the local optimum for $f_i$,

$\|x_i^0 - x^*\|^2 \le R_0^2, \ \forall i \in [N], \quad \|x_i^\star - x^*\|^2 \le R_*^2, \ \forall i \in [N].$

The last two constraints are standard in the PEP literature. The PEP formulation above can be transformed into an *equivalent* semidefinite programming (SDP) problem, which is convex and can therefore be solved efficiently. As long as the condition $\dim(G) \le d$ holds, the *exact* worst-case optimization error after $K$ iterations of Algorithm 1 is equal to the optimal value of this SDP.

This equivalence implies that for the given function class, there exists a specific function on which Algorithm 1 achieves precisely the worst-case error predicted by the PEP. Moreover, no function in the class can lead to a larger error under the same algorithm. A more detailed derivation of this formulation, along with its final SDP representation, is provided in the *Appendix*.

To efficiently solve the resulting SDPs arising from the PEP formulation, we utilize the JuMP modeling language in Julia. JuMP (Lubin et al., 2023) offers a flexible and high-level interface for formulating convex optimization problems, including SDPs, and facilitates seamless integration with state-of-the-art solvers. For this work, we employ the MOSEK solver (ApS, 2019), which is well-suited for large-scale and structured convex optimization problems, particularly semidefinite and conic programs. This combination enables us to model and solve the PEP instances accurately and efficiently. All experiments are conducted on a computing platform equipped with dual AMD EPYC 9654 processors (each with 96 cores) and 256 GB of RAM.

Using this approach, we rigorously compare the performance of the proposed Algorithm 1 with its centralized counterpart, i.e., the update rule in (2). Without loss of generality, we first consider a dataset composed of $N = 2$ types of data samples. In this case, we have $f(x) = \frac{1}{2}\sum_{i=1}^2 f_i(x)$. We assume that the data distributions are heterogeneous and hence $f_1$ and $f_2$ have different smoothness properties (e.g., $L_1 = \frac{1}{3}$ and $L_2 = 3$). The results are summarized in Fig. 1(a). It is evident that Algorithm 1 can significantly reduce the number of iterations required to achieve a given accuracy level, compared to its centralized counterpart (denoted as "GD"). It is worth noting that although we leverage local geometry—captured by the local smoothness constants $L_i$—to accelerate convergence, directly applying this strategy to existing decentralized gradient methods is not viable, as it leads to a steady-state error (see the result represented by the black curve in Fig. 1(a)). This highlights the necessity and significance of our proposed design. Moreover, to

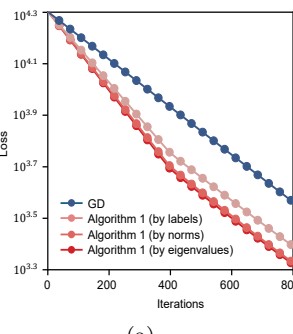 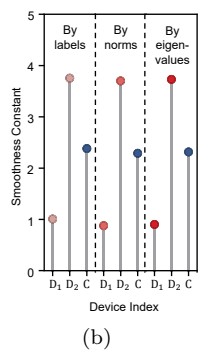 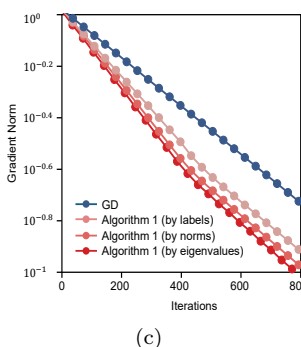

|          (a)          |          (b)          |          (c)          |

Figure 2: Comparison of Algorithm 1 with its centralized counterpart (labeled GD) using the W8A dataset. (a) and (c) compare the convergence performance between the centralized approach (labeled GD) and our proposed decentralized method under three partitioning schemes for the entire data, labeled Algorithm 1 (by labels), Algorithm 1 (by norms), and Algorithm 1 (by eigenvalues), respectively. The results show that in all data partitioning schemes, using decentralization substantially reduces the number of iterations required to reach a certain accuracy level. (b) plots the resulting smoothness constants under the three partitioning schemes, respectively, where on the x-axis, "C" represents the smoothness constant of the entire dataset (used in the centralized case), and "$D_1$" and "$D_2$" represent the smoothness constants for device 1 and device 2, respectively.

assess the impact of switching timing in Algorithm 1, we present results for three different switching points, represented by the three curves with varying shades of red in Fig. 1(a). In all cases, Algorithm 1 achieves faster convergence than the centralized counterpart, demonstrating that its performance is not sensitive to the choice of switching timing.

**Theorem 1.** *(Convergence Acceleration of Algorithm 1) Consider the optimization problem defined in (1), where the function $f$ belongs to the class $\mathcal{F}_{\mu,L}$, characterized by the smoothness parameter $L$ and strong convexity parameter $\mu$. Let $\mathcal{A}_1$ denote Algorithm 1 with local step sizes $\frac{1}{L_i}$, and $\mathcal{A}_c$ denote centralized gradient descent with a uniform step size $\frac{1}{L}$. For the performance metric*

$$P_{\mathcal{A}} = \max_{f \in \mathcal{F}_{\mu,L}} \left\{ \frac{1}{N} \sum_{i=1}^{N} \|x_{i,\mathcal{A}}^k - x^*\|^2 \right\},$$

*which measures the worst-case optimization error at iteration $k$ over all $f \in \mathcal{F}_{\mu,L}$, we consider the ratio $P_{\mathcal{A}_1}/P_{\mathcal{A}_c}$ as the relative acceleration of Algorithm 1 compared with centralized gradient descent (with values below 1 indicating acceleration and smaller ratios indicating stronger acceleration). Figure 1(b) gives the exact ratio values under different function classes, and the ratio values being strictly less than 1 for all tested function classes $\mathcal{F}_{\mu,L}$ in the heterogeneous setting confirm that Algorithm 1 with local step sizes achieves a strictly faster convergence rate than centralized gradient descent under all considered function classes.*

*Proof.* First, we have $P_{\mathcal{A}} < \infty$ for any finite $k$, which implies that the worst-case optimization error of Algorithm 1 over the function class $\mathcal{F}_{\mu,L}$ is always finite for any $k$. Therefore, the set $\arg\max_{f \in \mathcal{F}_{\mu,L}} \left\{ \frac{1}{N} \sum_{i=1}^{N} \|x_{i,\mathcal{A}}^k - x^*\|^2 \right\}$ is guaranteed to be nonempty. This means that there exists at least one function instance $f_{\mathcal{A}}^*$ such that, Algorithm 1, when executed with step-size scheme $\mathcal{A}$, indeed attains its worst-case performance at iteration $k$.

For any such maximizer $f_{\mathcal{A}}^* \in \arg\max_{f \in \mathcal{F}_{\mu,L}} \left\{ \frac{1}{N} \sum_{i=1}^{N} \|x_{i,\mathcal{A}}^k - x^*\|^2 \right\}$, the value $P_{\mathcal{A}}$ returned by the PEP is therefore an *exact*, not merely upper-bounding, characterization of the true worst-case error over $\mathcal{F}_{\mu,L}$. In other words, the PEP does not produce a loose bound: it identifies a specific problem instance on which Algorithm 1 performs exactly as poorly as the bound indicates. With this interpretation, the ratio $\frac{P_{\mathcal{A}_1}}{P_{\mathcal{A}_c}}$ directly compares the exact worst-case optimization errors of Algorithm 1 versus centralized gradient descent.

A value $\frac{P_{A_1}}{P_{A_c}} < 1$ therefore means that—in the worst case—Algorithm 1 has strictly smaller optimization error than centralized gradient descent at iteration $k$. Hence, the ratio values in Figure 1(b) quantitatively capture the exact speed gains achieved by Algorithm 1. □

To examine how the difference between the smoothness constants $L_1$ and $L_2$ influences the acceleration achieved by Algorithm 1 over its centralized counterpart, we compute the ratio of the optimization error of Algorithm 1 to that of the centralized method across various values of the smoothness constants. As presented in Theorem 1, a ratio smaller than one clearly indicates that Algorithm 1 outperforms its centralized counterpart, with smaller ratios corresponding to greater acceleration. Fig. 1(b) presents the comparison results when Algorithm 1 is under different $(L_1, L_2)$ pairs with a fixed sum. Note that for all such pairs, the centralized counterpart has a fixed smoothness constant of $L = \frac{L_1 + L_2}{2} = 1$. The results indicate that a larger difference between the smoothness constants—which corresponds to greater heterogeneity in the data distributions across the two devices—yields greater acceleration of Algorithm 1 over its centralized counterpart. In Fig. 1(c), we evaluate the influence of the number of devices on this acceleration. The results clearly show that the acceleration remains essentially unchanged regardless of the number of devices involved.

## 4 Results

In this section, we present three experiments on benchmark datasets to validate the finding that decentralization can indeed reduce the number of iterations required to reach an optimal solution. The code used in this study can be found at[1]

### 4.1 Logistic regression based Web categorization

In this experiment, we consider a logistic regression categorization task using the W8A dataset (Chang & Lin, 2011). In the dataset, each entry contains one feature and one label. The feature is a vector of 300 sparse keyword attributes extracted from each web page, and the label indicates the category of the entry (Platt, 1999). We consider a logistic regression model for the data, where the local loss function are gived by the following function:

$$f_i(x) = \frac{1}{m} \sum_{j=1}^{m} \log \left( 1 + \exp(-b_j a_j^T x) \right) + \frac{\mu}{2} \|x\|_2^2, \tag{3}$$

where $a_j \in \mathbb{R}^{300}$ denotes the feature of the $j$th data entry, $b_j \in \{-1, 1\}$ denotes its label and $m$ denotes the number of data samples.

In the experiment, we compare our decentralized optimization approach to (3) (i.e., Algorithm 1) and its centralized counterpart (see update rule in (2)). Based on our results in Section 3.2, greater heterogeneity in data distributions leads to increased acceleration of Algorithm 1. Therefore, we consider three data partitioning schemes that produce heterogeneous data distributions. In the first partitioning scheme, we split the data into two groups based on their label values: all entries with label "1" are assigned to one device, while all entries with label "−1" are assigned to the other device. In the second partitioning scheme, we sort all data entries based on the norm of the feature vector $\|a_j\|$ and assign the smaller half to device 1, while the larger half is assigned to device 2. In the third partitioning scheme, we order the data entries according to the maximum eigenvalue of $a_j a_j^T$ and allocate the smaller half to device 1 and the remainder to device 2. Since the smoothness constants are determined by the Hessian matrix—which is directly influenced by the factors used in the three partitioning schemes—the two devices have different smoothness constants in all cases. In the decentralized case, we run a centralized gradient descent (labeled as GD) with the gradient calculated from all data entries without any partitioning. The comparison results are summarized in Fig. 2. It is clear that decentralization can indeed significantly enhance convergence speed.

---

[1]https://anonymous.4open.science/r/Acc-ML-via-Decentral-050D/README.md.

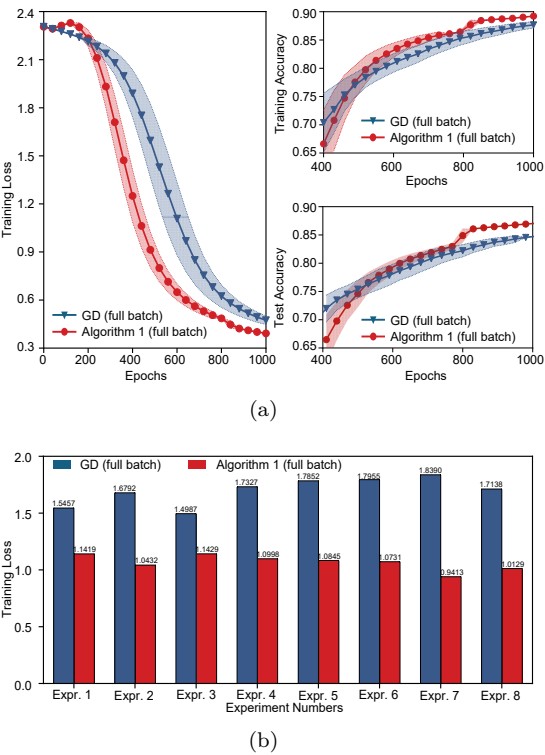
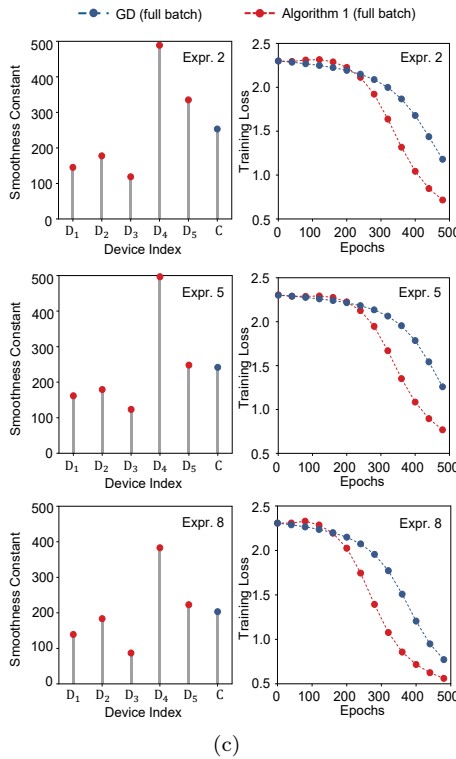

Figure 3: Comparison of Algorithm 1 with its centralized counterpart (labeled GD) using the MNIST dataset. In this experiment, both algorithms use full-batch gradient computation: the centralized approach processes the entire dataset, while in the decentralized approach, each device computes gradients using all the data allocated to it. (a) shows that the decentralized approach achieves faster convergence than the centralized counterpart in terms of training loss, training accuracy, and test accuracy. (b) compares the training loss over eight runs with different estimated smoothness constants and random initializations. (c) illustrates the estimated smoothness constants alongside the training loss trajectories of three selected runs from the eight. On the x-axis representing the device index, the label "C" denotes the centralized case, and "$D_1$" through "$D_5$" represent the devices in the decentralized case.

## 4.2 Handwritten digits classification on MNIST

In this experiment, we consider a neural network based classifier for the MNIST dataset of handwritten digits (LeCun et al., 1994), a widely used benchmark for training and evaluating models in the field of machine learning (Deng, 2012). The dataset contains 60,000 training images and 10,000 testing images, with each set containing a roughly equal number of images for each digit from 0 to 9. We employ a classifier based on a deep convolutional neural network (CNN). The architecture begins with two convolutional layers, with 16 and 32 filters respectively, each followed by a max pooling layer. Finally, the network includes a fully connected dense layer that maps the extracted features to 10 output classes. In the decentralized case, the data are partitioned across five devices in a label-based, heterogeneous manner. Specifically, all data entries with the same label are assigned to the same device, and each device holds data corresponding to two distinct labels. For each device, we estimate the smoothness constant using 1,000 data entries, following its formal definition. The results are summarized in Fig. 3. It is evident that decentralization can substantially reduce the number of iterations required to reach an optimal solution. It is worth noting that the experiment was conducted multiple times using different smoothness constants, each estimated from distinct subsets of data (see Fig. 3(c) for three instances of estimated smoothness constants). Despite variations in these estimates, the results consistently indicate that decentralization yields significant acceleration over the centralized approach. Furthermore, in this experiment, we use the entire dataset (full batch) for gradient calculation,

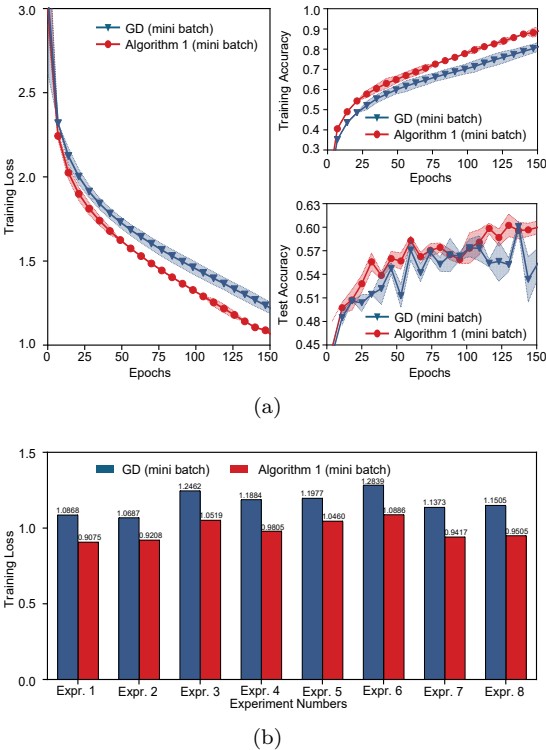

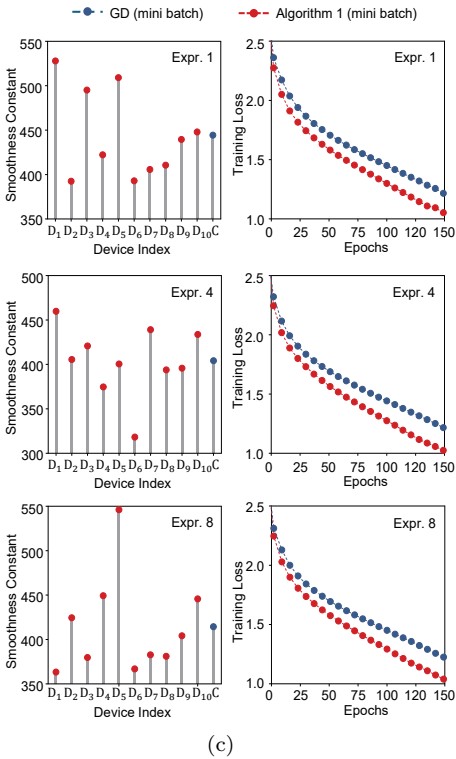

Figure 4: Comparison of Algorithm 1 with its centralized counterpart (labeled GD) using the CIFAR-10 dataset. In this experiment, both algorithms use mini-batch gradient computation: in each iteration, the centralized approach processes ten randomly selected samples, while each device in the decentralized approach computes gradients using one sample from the data allocated to it (so both approaches process the same number of ten samples in each iteration). (a) shows that the decentralized approach achieves faster convergence than the centralized counterpart in terms of training loss, training accuracy, and test accuracy. (b) compares the training loss over eight runs with different estimated smoothness constants and random initializations. (c) illustrates the estimated smoothness constants alongside the training loss trajectories of three selected runs from the eight. On the x-axis representing the device index, the label "C" denotes the centralized case, and "$D_1$" through "$D_{10}$" represent the devices in the decentralized case.

which may become computationally expensive for more complex images. Therefore, in the next experiment, where we consider a dataset with more complicated images, we employ mini-batch sampling to compute gradients at each iteration.

### 4.3 Image classification on CIFAR-10

In this experiment, we train a deep neural network on the CIFAR-10 dataset (Krizhevsky et al., 2009), which offers greater diversity and complexity compared to the MNIST dataset. The CIFAR-10 dataset consists of 60,000 color images of size $32 \times 32$ pixels, divided into 10 classes. Each class contains 6,000 images, with 5,000 allocated for training and 1,000 for testing. In this experiment, we employ a four-level CNN–based classifier. The architecture consists of four convolutional layers with 32, 64, 128, and 128 filters, respectively. Max pooling layers are applied after the second and fourth convolutional layers. Finally, the network includes a global average pooling layer and a fully connected dense layer that maps the extracted features to 10 output classes. In our implementation of the decentralized Algorithm 1, we partition the entire dataset across ten devices based on labels—that is, all data entries with the same label are assigned to the same device. Similar to the previous experiment, we used a small number of samples to estimate the smoothness constant for each device. However, instead of using full-batch gradient computation, we employ mini-batch based

gradient computation, which only uses a few training samples at an iteration. To ensure a fair comparison, in Algorithm 1, each device selects a single random sample for gradient computation in each iteration, while the centralized approach selects 10 random samples per iteration. In this way, both algorithms process the same total number of samples in each iteration. The results are summarized in Fig. 4. It is evident that decentralization can substantially reduce the number of iterations required to reach an optimal solution. It is worth noting that the experiment was conducted multiple times using different smoothness constants, each estimated from distinct subsets of data (see Fig. 4(c) for three instances of estimated smoothness constants). Despite variations in these estimates, the results consistently demonstrate that decentralization achieves clear acceleration compared to the centralized case.

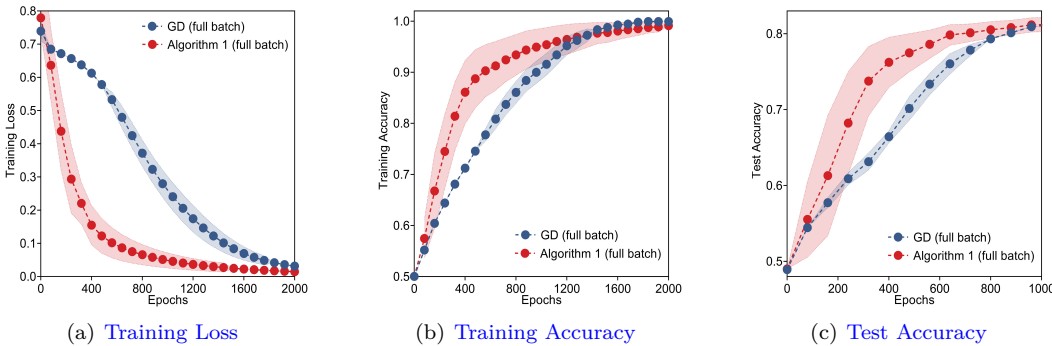

(a) Training Loss        (b) Training Accuracy        (c) Test Accuracy

Figure 5: Comparison of Algorithm 1 with its centralized counterpart (labeled GD) on the SST-2 dataset. In this experiment, both algorithms use full-batch gradient computation: the centralized approach processes the entire dataset, while in our decentralization based approach, each device computes gradients using all the data allocated to it. (a)-(c) show that our approach converges faster than the centralized counterpart.

## 4.4 Sentiment analysis on SST-2

In this experiment, we consider a BERT-based classifier for the Stanford Sentiment Treebank (SST-2) dataset (Socher et al., 2013), a widely used benchmark for evaluating language-model fine-tuning in the field of natural language processing (Wang et al., 2018). SST-2 is a binary sentiment classification dataset with positive and negative labels, containing a total of $67,349$ training samples. We partition the training dataset across five devices using a Dirichlet distribution with parameter 0.1, creating a high degree of label-based heterogeneity among the devices. Our classifier employs a BERT-base encoder with a classification head (Devlin et al., 2019). The BERT layers are frozen except for the last encoder layer, which is fine-tuned along with the classification head to adapt the representations to the SST-2 task. Both centralized gradient descent (GD) and our proposed Algorithm 1 are evaluated under this setup. Since the labels of the test dataset are not publicly available, we evaluate the test accuracy on the validation dataset instead. The results are summarized in Fig. 5. It is evident that our approach outperforms the centralized baseline, achieving accelerated convergence in language-model fine-tuning.

*Remark* 1. In logistic regression experiments, the smoothness constant $L_i$ for each client is computed directly from the local Hessian matrix using its largest eigenvalue. In neural network training, including CNN training and BERT fine-tuning, since a closed-form expression of $L_i$ is generally unavailable, we estimate it numerically using a gradient difference approximation. Specifically, a mini-batch of local data is fixed and multiple model parameters $x$ are random sampled from a region that empirically covers typical parameter variations during training. For each sampled parameter, a small perturbation $\varepsilon$ is applied and the ratio of gradient difference to parameter difference is computed. This procedure is repeated tens of thousands of times, and then the maximal value of the resulting ratios is taken as an estimate of the local smoothness constant $L_i$.

## 5 Discussion and conclusion

Our finding—that decentralization can accelerate the convergence of optimization algorithms—is surprising, but it carries significant implications. Faster convergence, meaning fewer iterations to reach a satisfactory solution, is critical for meeting the timing constraints of real-time systems such as autonomous vehicles. Additionally, reducing computational complexity is essential when training large-scale neural networks. Our finding provides strong support for the development of decentralized optimization algorithms—not only as a workaround when centralized approaches are infeasible due to data aggregation constraints imposed by privacy regulations, but also as a more powerful alternative when convergence speed is critical. Notably, our result was obtained under the assumption that the centralized server possesses computational power equal to the combined capabilities of all devices in the decentralized setup. This already gives the centralized approach an advantage in comparison. In practice, however, decentralized systems can harness the aggregate computational power of multiple devices, while a single centralized server remains inherently limited. A limitation of the current study is that it does not account for communication overhead, which may become non-negligible in settings where high-speed communication channels are unavailable. In fact, when data are distributed across multiple locations, the communication overhead required to aggregate raw data for centralized optimization is typically much greater than that incurred by exchanging optimization variables in decentralized approaches. Therefore, we expect that a similar conclusion holds in this case and plan to systematically investigate this problem in future work.

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

## A  Performance evaluation

It is worth noting that although numerous optimization algorithms have been proposed, quantifying and comparing their performance—typically evaluated through worst-case guarantees in the optimization and machine learning communities (Memirovski & Yudin, 1983; Polyak, 1987; Nesterov, 2013)—remains intrinsically challenging. This difficulty arises because most existing results, which are primarily based on analytical methods, tend to be overly conservative: they generally provide only sufficient (but not necessary) conditions for convergence, or offer asymptotic rates of convergence rather than exact quantitative characterizations (Taylor et al., 2017b). Recently, a computational approach was proposed that can provide the *exact* worst-case performance of an optimization algorithm. This approach is based on formulating performance analysis as a semidefinite program (see, e.g., Vandenberghe & Boyd (1996)) and is also known as the performance estimation problem (PEP) (Drori & Teboulle, 2014; Taylor et al., 2017b). This approach can obtain exact performance bounds for algorithms over specified function classes (Taylor et al., 2017b), in contrast to traditional asymptotic analyses that often produce loose guarantees. This approach has recently been extended to the decentralized case (Colla & Hendrickx, 2023).

The idea behind the PEP is to frame the worst-case performance evaluation of an algorithm as an optimization problem over all admissible functions (e.g., smooth and convex) and initial conditions. Although this optimization is inherently infinite-dimensional, as it involves a continuous function over its variables, Taylor et al. (2017a;b) demonstrate that it can be solved *exactly* as a finite-dimensional convex optimization problem by replacing the function constraint with an equivalent finite dimensional interpolation condition.

Generally, the framework can be formulated as follows:

$$\sup_{f,x^0,\ldots,x^K,x^*} \mathcal{P}(\mathcal{O}_f, x^0, \ldots, x^K, x^*) \tag{PEP}$$

$$\text{such that} \quad f \in \mathcal{F},$$
$$x^* \text{ is optimal for } f,$$
$$x^1, \ldots, x^K \text{ are generated from } x^0 \text{ by method } \mathcal{M} \text{ with } \mathcal{O}_f,$$
$$\|x^0 - x^*\|_2 \leq R,$$

where $\mathcal{M}$ denotes an optimization algorithm, $\mathcal{O}_f$ denotes the oracle that returns the corresponding function value and gradient at a given point of the function $f$, $\mathcal{F}$ is the class of functions to which $f$ belongs (e.g., strongly convex functions or general convex functions), $K$ is the number of steps the algorithm $\mathcal{M}$ performs, and $R$ is a given constant. By solving the PEP, one can obtain *exact* performance bounds for algorithms over designated function classes, in contrast to traditional asymptotic analyses, which often yield loose guarantees.

The PEP framework can be directly applied to evaluate the performance of centralized gradient descent algorithms. To evaluate the performance of our proposed decentralized approach, we extend the result in Colla & Hendrickx (2023) for homogeneous step sizes to the heterogeneous step size case and provide a new PEP formulation.

To describe the PEP framework, we first define the augmented states as $x^k = [x_1^k, \ x_2^k, \ \ldots, \ x_N^k] \in \mathbb{R}^{d \times N}$, where the superscript $k$ belongs to the index set $I_K = \{0, \ldots, K\}$. Similarly, we define the augmented local optimum and global optimum as $x^\star = [x_1^\star, \ x_2^\star, \ \ldots, \ x_N^\star]$ and $x^* = [x_1^*, \ x_2^*, \ \ldots, \ x_N^*]$, respectively (where $x_1^* = x_2^* = \cdots = x_N^*$ will be enforced in the PEP formulation). To simplify notation, we extend the index set $I_K$ for augmented states to include these two optimal states, defining the augmented index set as $I_K^{\star,*} = \{0, \ldots, K, \star, *\}$.

In a similar manner, we define the augmented gradients and function values as

$$g^k = [g_1^k, \ g_2^k, \ \ldots, \ g_N^k] \in \mathbb{R}^{d \times N}, \quad f^k = [f_1^k, \ f_2^k, \ \ldots, \ f_N^k] \in \mathbb{R}^{1 \times N}, \quad \text{for } k \in I_K^{\star,*}.$$

Under these definitions, we have

$$P = [x^0 \quad g^0 \quad g^1 \quad \ldots \quad g^K \quad g^\star \quad g^* \quad x^\star \quad x^*] \in \mathbb{R}^{d \times [(K+6)N]},$$
$$G = P^T P \in \mathbb{S}_+^{((K+6)N) \times ((K+6)N)}, \ F = [f^0 \quad f^1 \quad \ldots \quad f^K \quad f^\star \quad f^*] \in \mathbb{R}^{1 \times (K+3)N},$$

where $\mathbb{S}_+$ denotes the set of symmetric positive semidefinite matrices. We use the standard notation $e_i \in \mathbb{R}^d$ to denote the unit $d$-dimensional vector with a single 1 in the $i$-th component and $\mathbf{1}_N \in \mathbb{R}^N$ as the all one vector. The following vectors are employed to select specific columns in $P$ and $F$:

$$\mathbf{f}_i^k = e_{kN+i} \in \mathbb{R}^{(K+3)N}, \mathbf{g}_i^k = e_{(k+1)N+i} \in \mathbb{R}^{(K+6)N}, \mathbf{g}_i^* = e_{(K+3)N+i} \in \mathbb{R}^{(K+6)N},$$
$$\mathbf{g}_i^\star = e_{(K+2)N+i} \in \mathbb{R}^{(K+6)N}, \mathbf{x}_i^\star = e_{(K+4)N+i} \in \mathbb{R}^{(K+6)N},$$
$$\mathbf{f}_i^\star = e_{(K+1)N+i} \in \mathbb{R}^{(K+3)N}, \mathbf{f}_i^* = e_{(K+3)N+i} \in \mathbb{R}^{(K+3)N},$$
$$\mathbf{x}_i^0 = e_i \in \mathbb{R}^{(K+6)N}, \forall i \in [N], k \in I_K; \mathbf{x}^* = e_{(K+5)N+i} \in \mathbb{R}^{(K+6)N}.$$

The stacked version of these vectors is analogous to the formulation above. Now we are in the position to give our formulation of the PEP for the decentralized algorithm as follows

$$\max_{F,G} \quad F * \left( \sum_{i=1}^{N} \mathbf{f}_i^K - \mathbf{f}^* \mathbf{1}_N \right) \tag{4}$$

$$\text{subject to} \quad \langle G, A(\{\mathbf{x}_i^k, \mathbf{g}_i^k, \mathbf{f}_i^k\}_{k \in \{p,q\}}, \mu_i, L_i)\rangle \leq F * (\mathbf{f}_i^p - \mathbf{f}_i^q), \ \forall i \in [N], \ p, q \in I_K^{\star;*}, \tag{5}$$

$$\{\mathbf{x}_i^k\}_{i \in [N], k \in I_K} \text{ are generated recursively by Algorithm 1,} \tag{6}$$

$$\langle G, \mathbf{g}^* \mathbf{1}_N \mathbf{1}_N^T \mathbf{g}^{*T}\rangle = 0, \quad \langle G, \mathbf{g}_i^{\star} \mathbf{g}_i^{\star T}\rangle = 0, \ \forall i \in [N], \tag{7}$$

$$\langle G, (\mathbf{x}_i^0 - \mathbf{x}^*)(\mathbf{x}_i^0 - \mathbf{x}^*)^T\rangle \leq R_0^2, \ \forall i \in [N], \tag{8}$$

$$\langle G, (\mathbf{x}_i^{\star} - \mathbf{x}^*)(\mathbf{x}_i^{\star} - \mathbf{x}^*)^T\rangle \leq R_*^2, \ \forall i \in [N], \tag{9}$$

$$\langle G, (\mathbf{x}_1^0 - \mathbf{x}_i^0)(\mathbf{x}_1^0 - \mathbf{x}_i^0)^T\rangle = 0, \ \forall i \in [N], \tag{10}$$

$$\langle G, (\mathbf{x}_1^* - \mathbf{x}_i^*)(\mathbf{x}_1^* - \mathbf{x}_i^*)^T\rangle = 0, \ \forall i \in [N], \tag{11}$$

$$G \succeq 0, \tag{12}$$

$$\text{rank}(G) \leq d, \tag{13}$$

where

$$A(\{\mathbf{x}_i^k, \mathbf{g}_i^k, \mathbf{f}_i^k\}_{k \in \{p,q\}}, \mu_i, L_i)$$
$$= \frac{1}{2} \left[ (\mathbf{x}_i^p - \mathbf{x}_i^q)\mathbf{g}_i^{qT} + \mathbf{g}_i^q(\mathbf{x}_i^p - \mathbf{x}_i^q)^T \right]$$
$$- \frac{\mu_i(L_i - \mu_i)}{2L_i^2} \left[ (\mathbf{x}_i^q - \mathbf{x}_i^p)(\mathbf{g}_i^q - \mathbf{g}_i^p)^T + (\mathbf{g}_i^q - \mathbf{g}_i^p)(\mathbf{x}_i^q - \mathbf{x}_i^p)^T \right]$$
$$+ \frac{1}{2} \left(1 - \frac{\mu_i}{L_i}\right) \left[ \frac{1}{L_i}(\mathbf{g}_i^p - \mathbf{g}_i^q)(\mathbf{g}_i^p - \mathbf{g}_i^q)^T + \mu_i(\mathbf{x}_i^p - \mathbf{x}_i^q)(\mathbf{x}_i^p - \mathbf{x}_i^q)^T \right],$$

and $\langle \cdot, \cdot \rangle$ denotes the standard matrix inner product defined as $\langle A, B \rangle = \text{trace}(AB^T)$, $R_*$ bounds the distance between local and global optima, i.e., $\|x_i^{\star} - x^*\| \leq R_*$, $R_0$ bounds the distance between the initial point and the global optimum, i.e., $\|x_i^0 - x^*\| \leq R_0$, and generally $R_* \ll R_0$. We use the measure here because an equal start guarantees that the algorithm generates local states uniformly across all agents. Thus, we do not explicitly make use of $f(\bar{x})$ here. The constraints in (4)–(8) are standard in the formulation of the PEP (Taylor et al., 2017b). The constraints in (10) enforce an identical starting point, while the constraints in (11) ensure that the global optimum is consistent across all agents.

# B    Additional experimental results

## B.1    Experimental results on various heterogeneous data distributions

We have conducted additional experiments to evaluate the learning performance of our Algorithm 1 under different levels of data heterogeneity using the MNIST dataset. More specifically, the training dataset consists of 10 classes. For each class, we partitioned the data across 5 agents according to a Dirichlet distribution with parameters $\alpha \in \{0.1, 0.5, 1, 5\}$. Note that the parameter $\alpha$ measures the heterogeneity in data distribution among all agents. A smaller $\alpha$ leads to a higher degree of data heterogeneity. Both Algorithm 1 and its centralized counterpart (called GD) used full-batch gradient computation.

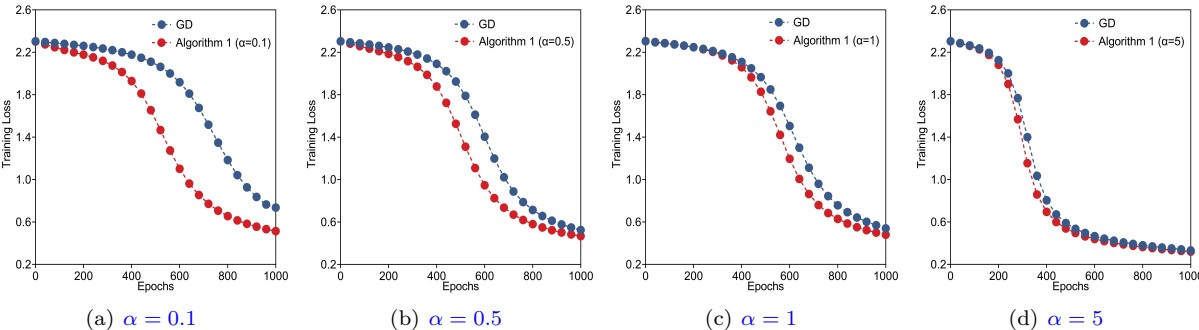

Figure 6: Training losses of Algorithm 1 and its centralized counterpart (labeled GD) for heterogeneous data distributions under different Dirichlet parameters $\alpha \in \{0.1, 0.5, 1, 5\}$ on the MNIST dataset.

Fig. 6 shows that a smaller Dirichlet parameter $\alpha$ (corresponding to a higher degree of heterogeneity in data distributions among agents) leads to more acceleration via decentralization.

## B.2 Experimental results on various network sizes

We have also conducted additional experiments to evaluate the efficacy of our Algorithm 1 under different network sizes (i.e., different number of devices). Specifically, we considered $N = 2, 5, 15, 25$ devices, respectively. The data are partitioned across agents according to a Dirichlet distribution with parameter $\alpha = 0.1$. All other parameters are the same as those in Section 4.2.

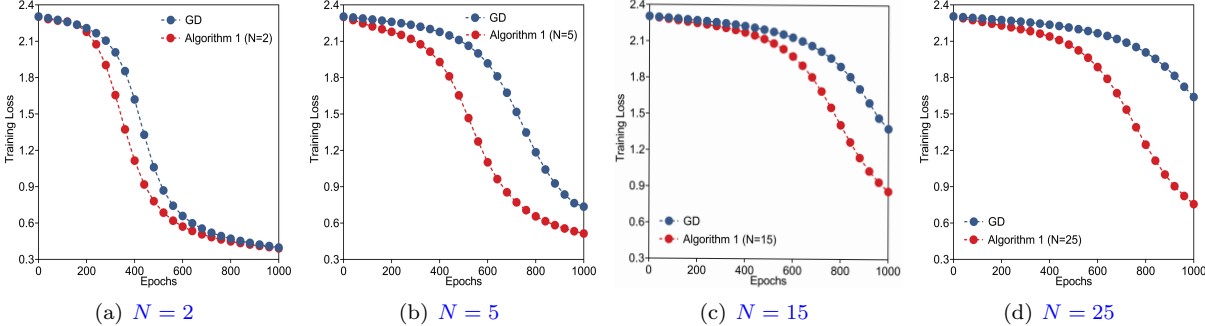

Figure 7: Training losses of Algorithm 1 and its centralized counterpart (labeled GD) for different network sizes $N, N \in \{2, 5, 15, 25\}$ on the MNIST dataset.

The results in Fig. 7 show that increasing the number of devices strengthens the performance advantage of our Algorithm 1 over its centralized counterpart.

## B.3 Comparison between Algorithm 1 with Decentralized GD and gradient tracking

To further evaluate the performance of Algorithm 1, we conducted additional experiments comparing it with decentralized gradient descent (labeled decentralized GD) from Lian et al. (2017) and gradient tracking from Pu et al. (2020) on the MNIST dataset. The data were partitioned across 5 devices according to a Dirichlet distribution with parameter $\alpha = 0.1$. For each device, we estimated the smoothness constant using $1,000$ data samples, based on its formal definition. The step sizes for decentralized GD and gradient tracking were set to the reciprocal of the average estimated smoothness constants. All algorithms used full-batch gradient computation.

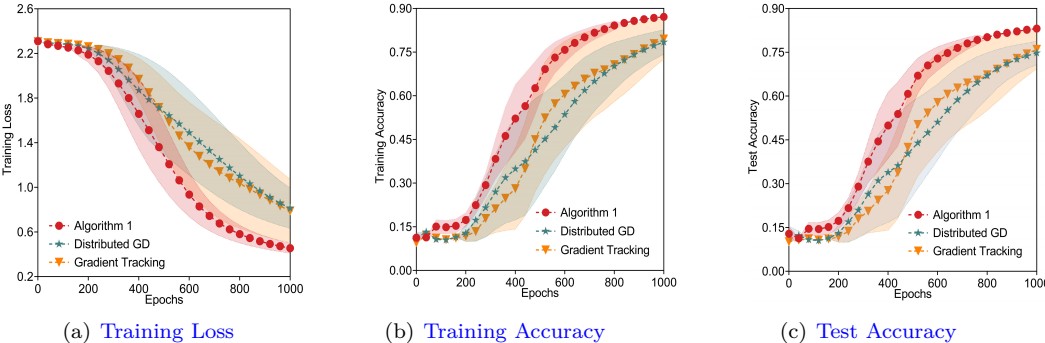

(a) Training Loss   (b) Training Accuracy   (c) Test Accuracy

Figure 8: Comparison of Algorithm 1 with decentralized GD in Lian et al. (2017) and gradient tracking in Pu et al. (2020) using the MNIST dataset. In this experiment, all algorithms use full-batch gradient computation, meaning that each device computes gradients using all the data allocated to it.

Fig. 8 shows that Algorithm 1 achieves faster convergence than both the decentralized GD and gradient tracking in terms of training loss, training accuracy, and test accuracy.

## B.4   Comparison of Algorithm 1 with FedOpt Variants and FedAvgM

To further evaluate the performance of Algorithm 1, we conducted additional experiments comparing it with FedOpt variants including FedAdam, FedYogi, and FedAdagrad from Reddi et al. (2021), as well as FedAvgM from Cheng et al. (2024), on the MNIST dataset. The data were partitioned across 5 devices, with each device containing samples from only two classes, corresponding to a highly heterogeneous (non-IID) setting. For each device, we estimated the smoothness constant using 1,000 data samples based on its formal definition. All algorithms employed full-batch gradient computation.

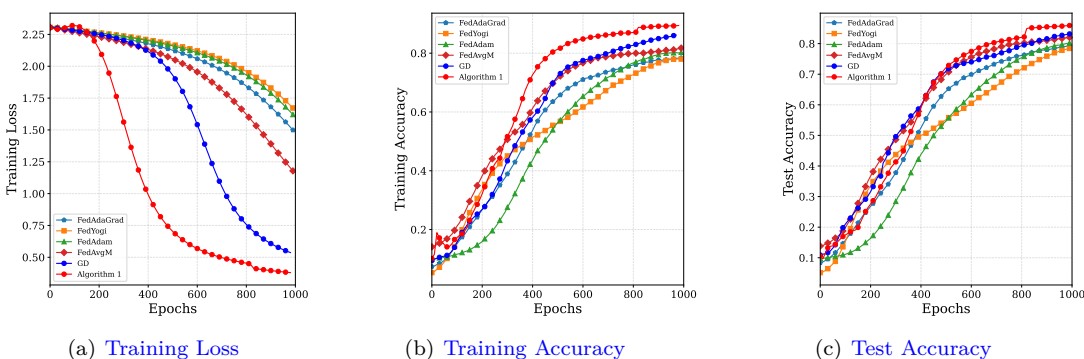

(a) Training Loss   (b) Training Accuracy   (c) Test Accuracy

Figure 9: Comparison of Algorithm 1 with FedOpt variants including FedAdam, FedYogi, and FedAdagrad from Reddi et al. (2021), as well as FedAvgM from Cheng et al. (2024). In this experiment, all algorithms use full-batch gradient computation, meaning that each device computes gradients using all the data allocated to it.

Fig. 9 shows that Algorithm 1 achieves faster convergence than FedOpt-based methods, including FedAdam, FedYogi, FedAdagrad, and FedAvgM, in terms of training loss, training accuracy, and test accuracy.

## B.5   Ablation Study of Algorithm 1 with Partial Agent Participation

We evaluate the proposed Algorithm 1 under a partial participation scheme, where at each communication round, some agents may be unable to participate due to failures in their local functions or communication delays. Specifically, at each communication round, each agent participates with probability

$p \in \{0.7, 0.8, 0.9, 1.0\}$, with $p = 1.0$ denoting participation in every round. The results shown in Fig. 10 demonstrate the robustness of our algorithm under partial participation in a 20-agent setting, where each agent possesses data from only two classes.

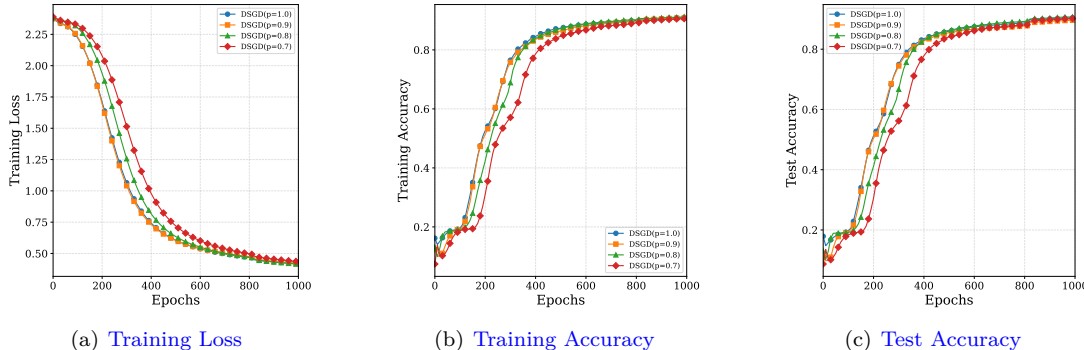

(a) Training Loss      (b) Training Accuracy      (c) Test Accuracy

Figure 10: Algorithm 1 with each agent having a participation probability of $p \in \{0.7, 0.8, 0.9, 1.0\}$ at each communication round on the MNIST dataset. In this experiment, all algorithms employ full-batch gradient computation, meaning that each device computes gradients using all locally allocated data.

The results demonstrate that the proposed Algorithm 1 remains robust under partial agent participation.

