# OpenReview forum: "Accelerating Optimization and Machine Learning through Decentralization"
_TMLR — Rejected by TMLR_

### Review · Reviewer_aZvh · 2026-02-14

**Summary Of Contributions:**

This paper challenges the long‑standing assumption that decentralized optimization is inherently slower than centralized optimization. The authors propose a server‑assisted decentralized gradient method (Algorithm 1) that uses heterogeneous local step sizes based on local smoothness constants, followed by a switch to a universal step size to eliminate steady‑state bias. Using the Performance Estimation Problem (PEP) framework, the paper provides worst‑case convergence guarantees and demonstrates acceleration over centralized gradient descent. Experiments on logistic regression (W8A), MNIST, and CIFAR‑10 further support the claim that decentralization can reduce the number of iterations needed to reach a target accuracy. While the paper appears to be novel and can stimulate new research, there are some shortcomings which should be addressed.

**Audience:**

Yes

**Audience Explanation:**

Yes, the paper is very relevant to the audience of TMLR. In fact, it falls under the category of decentralized algorithms in machine learning or federated learning which I believe is a quite important research direction.

**Broader Impact Concerns:**

I don’t think the paper has any ethical concerns.

**Claims And Evidence:**

Yes

**Claims Explanation:**

The core claim that decentralization can accelerate optimization by reducing the number of iterations required compared to centralized methods is supported by the clear theoretical proof with the PEP framework. However, the paper doesn’t report actual wall time or realistic computation/communication models. There is no empirical validation that a centralized full‑batch step and a multi‑device decentralized step actually have comparable latency on real hardware. These may be topics of future work.

**Requested Changes:**

While the paper is novel and well-written, I suggest the following changes:

-> The algorithm presented in sec 3.1 depends on the local smoothness constant L_i but no mention is made of how that is obtained. Clarifying that is important for the reader.

-> The results section does not include a demonstration using language models that use transformers. These networks are the most popular now and it is imperative to study the efficiency of this decentralized algorithm there. While adding a section with experimental results on LLMs is highly recommended, the authors can just mention how the algorithm will perform there if designing the experiment is too much of a heavylift.

->  Comparisons with other decentralized algorithms such as variants of decentralized SGD, gradient tracking, or other asynchronous methods are missing. Including that would be beneficial.

-> An important aspect that should be studied is fault-tolerance of the algorithm. If a decentralized agent fails due to some error or delays the broadcast of its state, the resiliency of the algorithm should be studied.

-> Comment regarding formatting - please ensure that the algorithm or figure corresponding to a particular section is placed after/before that section and not too far away. For e.g., fig 2 is on page 6 but actual mention of fig 2 is made on page 8. This makes it very inconvenient for the reader to follow along.

---

> ### Author Response · Authors · 2026-03-04
> **Response to Reviewer aZvh---Part I**
>
> $\rm\color{purple}{@aZvh\ @Response\ to\ Comment\ 1\  in\ Requested\ Changes:}$  Thank you for the suggestion. In the revised manuscript, we have clarified that in the logistic regression experiment, each client $i$'s smoothness constant $L_{i}$ is obtained by computing the maximal eigenvalue of the local Hessian matrix, while in neural network training, each client $i$'s smoothness constant $L_{i}$ is obtained using a gradient difference approximation. Specifically,
>
> **1) Logistic regression experiment.** The local objective function is convex and twice differentiable, given by
>
> $f_i(x) = \frac{1}{m} \sum_{j=1}^m \log\left(1 + \exp(-b_j a_j^\top x)\right) + \frac{\mu}{2}||x||_2^2$,
>
> where $a_j \in \mathbb{R}^{300}$ denotes the feature vector of the $j$-th sample, $b_j \in${$-1,1$} is its corresponding label, and $m$ is the number of local data samples. In this case, the smoothness constant is characterized by the spectral norm of the local Hessian matrix, which is given by $\nabla^2 f_i(x) = \frac{1}{m} \sum_{j=1}^m \left[ \frac{\exp(b_j a_j^\top x)}{(1 + \exp(b_j a_j^\top x))^2} \right] a_j a_j^\top + \mu I$. Noting that $\frac{\exp(b_j a_j^\top x)}{(1 + \exp(b_j a_j^\top x))^2} \le \frac{1}{4}$, where the maximum value $1/4$ is attained when $a_j^\top x = 0$ (e.g., $x=0$), we obtain the bound $\nabla^2 f_i(x) \preceq \frac{1}{4m} \sum_{j=1}^m a_j a_j^\top + \mu I$, where $A \preceq B$ means that $ B-A$ is positive semidefinite. Therefore, for client $i$, the exact Lipschitz constant is given by the largest eigenvalue of $\left(\frac{1}{4m} \sum_{j=1}^m a_j a_j^\top + \mu I\right)$. This quantity can be computed efficiently using standard numerical linear algebra routines.
>
> **2) Neural network training.** Since a closed-form expression of the smoothness constant is generally unavailable in neural network training, we estimate $L_i$ numerically using a gradient difference approximation. Concretely, for each client $i$,  we fix a mini-batch of local data and randomly sample  model parameter vectors $x$ from a region that empirically covers typical parameter variations during training. For each sampled point, we construct a perturbed parameter vector $x+\varepsilon$ with a small perturbation $\varepsilon>0$, and compute the ratio:
>
> $\frac{||\nabla f_{i}(x+\varepsilon)-\nabla f_i(x)||}{||(x+\varepsilon)-x||}.$
>
> This procedure is repeated over tens of thousands of times with different random initializations, and the largest ratio observed across all runs is taken as an estimate of the local smoothness constant. This numerical approximation effectively estimates the local Lipschitz constant of the gradient and is commonly adopted when closed-form expressions are not available for deep neural networks.
>
> To clarify these points, in the revised manuscript, we have added Remark 1 on page 10, which is presented as follows:
>
> **"Remark 1. In logistic regression experiments, the smoothness constant $L_i$ for each client is computed directly from the local Hessian matrix using its largest eigenvalue. In neural network training, including CNN training and BERT fine-tuning, since a closed-form expression of $L_i$ is generally unavailable, we estimate it numerically using a gradient difference approximation. Specifically, a mini-batch of local data is fixed and multiple model parameters $x$ are randomly sampled from a region that empirically covers typical parameter variations during training. For each sampled parameter, a small perturbation $\varepsilon$ is applied and the ratio of gradient difference to parameter difference is computed. This procedure is repeated tens of thousands of times, and then the maximal value of the resulting ratios is taken as an estimate of the local smoothness constant $L_i$."**
>
> Continued on Part II

---

> ### Author Response · Authors · 2026-03-04
> **Response to Reviewer aZvh---Part II**
>
> Continued from Part I
>
> $\rm\color{purple}{@aZvh\ @Response\ to\ Comment\ 2\ in\ Requested\ Changes:}$  We appreciate the reviewer for the suggestion. In the revised manuscript, we have conducted additional experiments on a Transformer-based model, namely BERT-base, using the SST-2 benchmark from the GLUE suite.
>
> Specifically, we consider a BERT-based classifier for the Stanford Sentiment Treebank (SST-2)
> dataset, which is a widely used benchmark for evaluating language-model fine-tuning in the field of natural language processing. The SST-2 dataset has positive and negative labels and contains a total of $67,349$ training samples. We partition the training
> dataset across five devices using a Dirichlet distribution with parameter $0.1$, creating a high degree of label-based heterogeneity among the devices. Our classifier employs a BERT-base encoder with a classification head. The BERT layers are frozen except for the last encoder layer, which is fine-tuned along with the classification head to adapt the representations to the SST-2 task. Both centralized gradient descent (GD) and our proposed Algorithm 1 are evaluated under this setup. Since the labels of the test dataset are not publicly available, we evaluate the test accuracy on the validation dataset instead.
>
> The experimental results are summarized in Fig. 5 on page 10 in the revised manuscript. Fig. 5 shows that our decentralized approach outperforms the centralized baseline, achieving accelerated convergence in language model fine-tuning. In addition, we explain that although we did not conduct experiments on very large language models due to computational constraints, our results on the BERT-based classifier demonstrate that the proposed algorithm is effective in Transformer-based architectures. We expect similar efficiency gains to hold for larger LLMs, as the algorithm primarily relies on decentralized gradient updates, which scale naturally with model size.
>
> $\rm\color{purple}{@aZvh\ @Response\ to\ Comment\ 3\  in\ Requested\ Changes:}$ Thank you for the suggestion. Following the reviewer's suggestion, in the revised manuscript, we have conducted additional experiments to compare the learning performance between Algorithm 1 and decentralized methods, including distributed gradient descent (called distributed GD)  in Lian et al. (2017) and gradient tracking
> in Pu et al. (2020), using the MNIST dataset.
>
> The experimental results are summarized in Fig. 8 on page 18. Fig. 8 shows that Algorithm 1 achieves faster convergence than both the distributed GD and gradient tracking in terms of training loss, training accuracy, and test accuracy.
>
> $\rm\color{purple}{@aZvh\ @Response\ to\ Comment\ 4\ in\ Requested\ Changes:}$ Thank you for this valuable suggestion. In the revised version, we evaluate the proposed Algorithm 1 under a partial participation scheme, where at each communication round, some agents may be unable to participate due to failures in their local functions or communication delays. The results shown in the newly added Fig. 10 on page 19 demonstrate the robustness of our algorithm under partial participation in a 20-agent setting, where each agent possesses data from only two classes. Fig. 10 shows that Algorithm 1 stays stable under partial agent participation in terms of training loss, training accuracy, and test accuracy.
>
> $\rm\color{purple}{@aZvh\ @Response\ to\ Comment\ 5\ in\ Requested\ Changes:}$ Thank you for pointing this out. We agree that placing figures and algorithms far from their first reference can interrupt the flow of reading and make the manuscript harder to follow. We have revised the formatting to ensure that each figure and algorithm now appears close to its first mention in the corresponding section. In particular, Figure 2 has been relocated to appear near the page where it is first discussed. We appreciate this helpful suggestion, which has improved the clarity and readability of the manuscript.

---

### Review · Reviewer_4mwn · 2026-02-16

**Summary Of Contributions:**

In this paper, the authors propose an improved algorithm based on PSGD, where each client computes gradients on its local data and a central server aggregates them to update the global model and broadcasts it back to the clients. The key idea is to turn cross-client heterogeneity in smoothness constants into faster convergence. Specifically, the method uses client-specific weight (step-size) coefficients $\alpha_i=1/L_i$ in the early training rounds to accelerate progress, and then switches to a shared coefficient $\alpha = 1/L$ near convergence to mitigate steady-state error induced by persistent heterogeneous weighting. In the numerical experiments (Section 4), the authors empirically show that, beyond convex logistic regression, the proposed approach can achieve faster convergence for non-convex CNN training on MNIST and CIFAR-10 under heterogeneous data partitions.

**Audience:**

Yes

**Audience Explanation:**

The idea of leveraging client heterogeneity to accelerate convergence is intriguing and could attract interest from the federated/decentralized learning community.

**Broader Impact Concerns:**

The idea of leveraging client heterogeneity to accelerate convergence is intriguing and could attract interest from the federated/decentralized learning community. However, the current evidence seems insufficient to support the breadth of the implied impact: key practical details (e.g., how $L_i$ is obtained and how robust the method is to estimation error and switching choices) are unclear, and the empirical evaluation is limited in scale and baselines. As a result, the paper may overstate how broadly the “heterogeneity helps” message applies without stronger theoretical and experimental support.

**Claims And Evidence:**

Yes

**Claims Explanation:**

Partially yes, but the evidence is not sufficient given the following concern.

**(W1) Missing or unclear theoretical support for the effectiveness of the proposed method.**

A central message of the paper—illustrated empirically in Fig. 1—is that larger cross-node heterogeneity (e.g., greater disparity among $L_i$) yields a smaller error ratio and thus faster convergence. I expected a clear theoretical support for these empirical results; however, I could not find one. The authors argue that standard convergence analyses in federated learning can be loose and conservative, and I agree with this point. However, even with the PEP framework, it remains unclear what concrete theoretical conclusions are actually established beyond numerical worst-case evaluations. In particular, I expected the paper to articulate (and ideally formalize) results along the lines of: (i) exploiting heterogeneous smoothness (client-specific $L_i$) is provably beneficial at least up to some neighborhood of the optimum, and (ii) switching to a common weighting based on $L$ near convergence is theoretically supported, including sensitivity to the switching point.

**(W2) Limited experimental evaluations and insufficient ablations.**

The current experimental section is relatively limited in scope, and several standard ablations and baselines are missing.

**(a)	Ablations on heterogeneity levels and number of clients.**

In federated learning, it is standard practice to control statistical heterogeneity using Dirichlet partitioning (via the concentration parameter $\alpha$) and to report performance as a function of both heterogeneity and the number of clients $N$.

**(b) Insufficient competitive baselines.**

The paper mainly compares against GD and a DGD ($\alpha_i = 1/L_i$), which is not sufficient to establish the advantage of the proposed method. More relevant baselines should be included, such as server-aggregation methods with momentum and adaptive updates (e.g., FedAvgM, and FedOpt variants such as FedAdam, FedYogi, and FedAdagrad). While these methods are often described in terms of model-update sharing, when the client's inner loop is restricted to a single step, they become closely related to gradient aggregation and should provide meaningful points of comparison.

**(c) Limited benchmarks/lack of larger-scale tasks.**
The experiments are restricted to small-scale settings (logistic regression and CNNs on MNIST/CIFAR-10). To better support the generality and practical relevance of the claims, the paper should include more challenging benchmarks, such as modern language-model fine-tuning tasks (e.g., GLUE). If the authors prefer to focus on convex objectives, one practical option is to freeze most network layers and train only the final layer (with weight decay), which yields a convex optimization problem while remaining realistic.

**(W3) Lipschitz-constant estimation in non-convex settings is under-specified.**

The proposed algorithm requires access to per-client smoothness constants $L_i$, but the manuscript does not clearly explain how these values are obtained or estimated in practice. This is particularly problematic for neural network objectives, where $L_i$ is typically unknown, can vary over the course of training, and is difficult to estimate reliably.

**Requested Changes:**

**(R1) Strengthen theoretical support for the effectiveness of the proposed method.**

The paper should provide a clearer and more convincing theory explaining why the proposed approach improves iteration complexity while avoiding bias/steady-state error.

(a) Why heterogeneity in client smoothness levels ($L_i$) can lead to faster convergence.

(b) Why switching to a common weight/step size $\alpha = 1/L$ near convergence mitigates bias.

(c) Sensitivity to the switching point.

**(R2) Expansion of experimental evaluations.**

(a) Add ablations over several heterogeneity levels and the number of clients $N$.

(b) Include more relevant baselines, e.g., FedAvgM and FedOpt variants such as FedAdam, FedYogi, and FedAdagrad with single inner loop to share gradients.

(c) Evaluate on additional benchmarks, ideally including modern language-model fine-tuning tasks (e.g., GLUE).

**(R3) Specify how per-client smoothness constants $L_i$ are obtained/estimated and assess robustness to estimation error.**

The proposed algorithm relies on per-client smoothness constants $L_i$, yet the manuscript does not clearly describe how these are computed/estimated in practice—especially for neural networks, where $L_i$ is typically unknown and difficult to estimate reliably when neural network models are employed.

---

> ### Author Response · Authors · 2026-03-04
> **Response to Reviewer 4mwn---Part I**
>
> $\rm\color{orange}{@4mwn\ @Response\ to\ Weakness\ (W1)\ and\ Requested\ Changes\ (R1):}$ Thank you for the comment.
> We emphasize that the PEP-based results are *formally valid and mathematically rigorous*, as they provide and compare the *exact* worst-case bounds for optimization algorithms over *a general class of functions* [R1], in contrast to traditional comparisons typically relying on loose upper bounds that can be overly conservative and sometimes misleading [R2]. It is also fundamentally different from numerical simulations on *specific function instances*. In the revised version, we clarify this point by formally summarizing our main theoretical results in Theorem 1. In this theorem, we rigorously characterize the performance of the proposed Algorithm 1 and compare it with centralized gradient descent. The theorem provides explicit proof of the acceleration of our proposed Algorithm 1, which is shown in Figure 1(b). Here we state Theorem 1 as follows for ease of reference:
>
> **"Theorem 1. (Convergence Acceleration of Algorithm 1)    Consider the optimization problem defined in (1), where the  function $f$ belongs to the class $\mathcal{F}_{\mu, L}$, characterized by the smoothness parameter $L$  and strong convexity parameter $\mu$. Let $\cal{A}_1$ denote Algorithm 1 with local step sizes $\frac{1}{L_i}$, and $\cal{A}_c$ denote centralized gradient descent with a uniform step size $\frac{1}{L}$. For the performance metric**
>
> $P\_{\\mathcal{A}}=\\max\_{f \\in \\mathcal{F}\_{\\mu,L}}\\{\\frac{1}{N}\\sum\_{i=1}^{N}||x\_{i,\\mathcal{A}}^{k} - x^{\\ast}||^2\\}$,
>
> **which measures the worst-case optimization error at iteration $k$ over all $f\in\mathcal{F}_{\mu, L}$, we consider the ratio**  $P_{\mathcal{A}1}$/$P_{\mathcal{A}c}$ **as the relative acceleration of Algorithm 1 compared with centralized gradient descent (with values below $1$ indicating acceleration and smaller ratios indicating stronger acceleration). Figure 1(b) gives the exact ratio values under different function classes, and the ratio values being strictly less than $1$ for all tested function classes $\mathcal{F}_{\mu, L}$ in the heterogeneous setting confirm that Algorithm 1 with local step sizes achieves a strictly faster convergence rate than centralized gradient descent under all considered function classes."**
>
> ***"Proof.*** **First, we have $P_{\mathcal{A}} < \infty$ for any finite $k$, which implies that the worst-case optimization error of Algorithm 1 over the function class $\mathcal{F}_{\mu, L}$ is always finite for any $k$.**
>
> **Therefore, the set** $\\arg\\max\_{f\\in\\mathcal{F}\_{\\mu, L}}\\{\\frac{1}{N}\\sum\_{i=1}^{N}||x\_{i,\\mathcal{A}}^{k} - x^*||^2\\}$ **is guaranteed to be nonempty. This means that there exists at least one function instance** $f_{\mathcal{A}}^{\ast}$ **such that,  Algorithm 1, when executed with step-size scheme** $\mathcal{A}$, **indeed attains its worst-case performance at iteration $k$.**
>
> **For any such maximizer**   $f\_{\\mathcal{A}}^{\\ast}\\in\\arg\\max\_{f\\in\\mathcal{F}\_{\\mu, L}}\\{\\frac{1}{N}\\sum\_{i=1}^{N}||x\_{i,\\mathcal{A}}^{k} - x^*||^2\\}$, **the value** $P_{\cal{A}}$ **returned by the PEP is therefore an *exact*, not merely upper-bounding, characterization of the true worst-case error over** $\mathcal{F}_{\mu, L}$. **In other words, the PEP does not produce a loose bound: it identifies a specific problem instance on which Algorithm 1 performs exactly as poorly as the bound indicates.**
>
> **With this interpretation, the ratio** $\frac{P_{\mathcal{A}1}}{P_{\mathcal{A}c}}$ **directly compares the exact worst-case optimization errors of Algorithm 1 versus centralized gradient descent.** **A value** $\frac{P_{\mathcal{A}1}}{P_{\mathcal{A}c}}<1$ **therefore means that—in the worst case—Algorithm 1 has strictly smaller optimization error than centralized gradient descent at iteration $k$. Hence, the ratio values in Figure 1(b) quantitatively capture the exact speed gains achieved by Algorithm 1."**
>
> [R1] Adrien B Taylor, Julien M Hendrickx, and François Glineur. Exact worst-case performance of first-order methods for composite convex optimization. *SIAM Journal on Optimization*, 27(3):1283–1313, 2017.
>
> [R2] Erwan Meunier and Julien M Hendrickx.  Several performance bounds on decentralized online optimization are highly conservative and potentially misleading. *IEEE 64th Conference on Decision and Control (CDC)*, IEEE, pp. 6201-6207, 2025.
>
> Continued on Part II

---

> ### Author Response · Authors · 2026-03-04
> **Response to Reviewer 4mwn---Part II**
>
> Continued from Part I
>
> $\rm\color{orange}{@4mwn\ @Response\ to\ Weakness\ (W1)\ and\ Requested\ Changes\ (R1)-(a):}$ Thank you for the suggestion.
> We first present a simple quadratic example borrowed from [R3] to provide an intuitive explanation of why, in the presence of heterogeneity in client smoothness levels, using local stepsizes can lead to faster convergence. We then show that our PEP-based framework provides the first rigorous theoretical evidence supporting this phenomenon.
>
> **An example of quadratic functions [R3]** For a parameter $a > 0$, consider the finite-sum optimization problem $\min_{x \in \mathbb{R}} f(x) := \frac{1}{2} \sum_{i=1}^{2} f_i(x)$ in the interpolation regime, where $f_1(x) = \frac{a}{2} x^2$ and $f_2(x) = \frac{1}{2} x^2$. If we solve this problem using gradient descent, $x_{t+1} = x_t - \frac{\alpha}{2} \big( \nabla f_1(x_t) + \nabla f_2(x_t) \big)$, and choose the step size $\alpha = \frac{1}{L}$, where $L = \frac{1+a}{2}$, the iteration complexity is $\Omega\left(a \log \frac{1}{\varepsilon}\right)$. In contrast, consider Algorithm 1 given by $x_{t+1} = x_t - \frac{1}{2} \big( \alpha_1 \nabla f_1(x_t) + \alpha_2 \nabla f_2(x_t) \big)$. For step sizes $\alpha_1 = \frac{1}{a}$ and $\alpha_2 = 1$, the iteration complexity becomes $\mathcal{O}\left(\log \frac{1}{\varepsilon}\right)$, which can be arbitrarily better than $\Omega\left(a \log \frac{1}{\varepsilon}\right)$ as $a \to \infty$.
>
> Although the phenomenon that heterogeneous smoothness levels can accelerate convergence when local stepsizes are used has been reported in the literature for *certain specific functions*, it has not been rigorously studied for a general class of functions. More importantly, existing discussions focus exclusively on the setting where the local optima of all clients coincide with the global optimum, since otherwise the use of local stepsizes will introduce steady-state errors.
> In this paper, we leverage the PEP framework to rigorously establish this phenomenon for a *general class of functions*. Furthermore, we propose an algorithm that systematically exploits smoothness heterogeneity to accelerate convergence while ensuring optimality, *even when the local optima of individual clients do not coincide with the global optimum*.
>
> Specifically, leveraging PEP, we compare the performance of algorithms using the exact optimization error, in contrast to traditional comparisons typically relying on loose upper bounds that can be overly conservative and sometimes misleading [R2]. Because our PEP-based approach produces an *exact* bound for a *general class of functions*, it identifies a specific problem instance on which Algorithm 1 performs exactly as poorly as the bound indicates. With this interpretation, the speed gain ratio in Figure 1 directly compares the exact worst-case optimization errors of Algorithm 1 with centralized gradient descent. A speed gain ratio less than one, therefore, means that—in the worst case—using local step sizes leads to a strictly smaller optimization error.  To the best of our knowledge, such results cannot be derived using traditional analytical techniques for general smooth and strongly convex functions.
>
> [R2] Erwan Meunier and Julien M. Hendrickx. Several Performance Bounds on Decentralized Online Optimization are Highly Conservative and Potentially Misleading. *IEEE 64th Conference on Decision and Control (CDC)*, IEEE, pp. 6201-6207, 2025.
>
> [R3] Mukherjee, Sohom, Nicolas Loizou, and Sebastian U. Stich.  Locally Adaptive Federated Learning, *arXiv:2307.06306*, 2023.
>
>
> $\rm\color{orange}{@4mwn\ @Response\ to\ Weakness\ (W1)\ and\ Requested\ Changes\ (R1)-(b):}$ Thank you for the comment.
> Note that when using a common stepsize ($\alpha$) after switching, the iteration in Algorithm 1 reduces to
>
> $x_{t+1} = x_t - \frac{\alpha}{N} \sum_{i=1}^N \nabla f_i(x_t) = x_t - \alpha \nabla f(x_t)$,
>
> which coincides exactly with the standard centralized gradient descent method applied to $f(x)$. Therefore, by using switching, the convergence behavior matches that of classical gradient descent, and the bias can thus be mitigated.
>
> $\rm\color{orange}{@4mwn\ @Response\ to\ Weakness\ (W1)\ and\ Requested\ Changes\ (R1)-(c):}$ Thank you for the comment. We include the sensitivity analysis of the switching point in Fig. 1(a), where the switch time is varied at different iteration steps. It can be seen that our algorithm remains stable and achieves higher efficiency under different switching timings. Moreover, since the server can readily monitor the global loss, it can easily determine the switching time by detecting when the loss begins to plateau. In all experiments presented in this paper, a monitoring mechanism is employed to determine the switch when the loss plateaus (see switching threshold $\epsilon$ in Algorithm 1), and the results consistently demonstrate the effectiveness of Algorithm 1.
>
> Continued on Part III

---

> ### Author Response · Authors · 2026-03-04
> **Response to Reviewer 4mwn---Part III**
>
> Continued from Part II
>
> $\rm\color{orange}{@4mwn\ @Response\ to\ Weakness\ (W2)-(a)\ and\ Requested\ Changes\ (R2)-(a):}$ Thank you for the comment. In the revised manuscript, we have conducted additional experiments to evaluate the learning performance of our Algorithm 1 under different levels of data heterogeneity using the MNIST dataset. More specifically, the training dataset consists of $10$ classes. For each class, we partition the data across $5$ agents according to a Dirichlet distribution with parameters $\alpha, \alpha\in${$0.1,0.5,1,5$}. Note that the parameter $\alpha$ measures the heterogeneity in data distributions among all agents. A smaller $\alpha$ leads to a higher degree of data heterogeneity. Both Algorithm 1 and its centralized counterpart (called GD) use full-batch gradient computation. The experimental results are summarized in the newly added Fig. 6 on page 17 in the revised manuscript.
>
> Fig. 6 shows that a smaller Dirichlet parameter $\alpha$ (corresponding to a higher degree of heterogeneity in data distributions among agents) leads to more acceleration via decentralization.
>
> Furthermore, following the reviewer's suggestion, we have also conducted additional experiments to evaluate the efficacy of our Algorithm 1 under different network sizes (i.e., different numbers of devices). Specifically, we considered $N=2,5,15,25$ devices, respectively. The data are partitioned across agents according to a Dirichlet distribution with parameter $\alpha = 0.1$. All remaining parameters are the same as those in Section 4.2. The experimental results are summarized in the newly added Fig. 7 on page 17 in the revised manuscript.
>
> The results in Fig. 7 show that increasing the number of devices strengthens the performance advantage of our Algorithm 1 over its centralized counterpart.
>
> $\rm\color{orange}{@4mwn\ @Response\ to\ Weakness\ (W2)-(b)\ and\ Requested\ Changes\ (R2)-(b):}$ Thank you for the suggestion. Following the reviewer's suggestion, in the revised manuscript, we have conducted additional experiments to compare the learning performance of Algorithm 1 with decentralized gradient descent, gradient tracking,  and  FedOpt-based methods, including FedAdam, FedYogi, FedAdagrad, and FedAvgM, on the MNIST dataset. The experimental results are summarized in the newly added Fig. 8 and Fig. 9 on page 18 in the revised manuscript. Fig. 8 and Fig. 9 show that Algorithm 1 achieves faster convergence than decentralized GD, gradient tracking, and FedOpt-based methods, including FedAdam, FedYogi, FedAdagrad, and FedAvgM, in terms of training loss, training accuracy, and test accuracy.
>
> $\rm\color{orange}{@4mwn\ @Response\ to\ Weakness\ (W2)-(c)\ and\ Requested\ Changes\ (R2)-(c):}$ Thank you for the suggestion. In the revised manuscript, we have conducted additional experiments to evaluate the effectiveness of our approach on a modern language-model fine-tuning benchmark, namely SST-2 from the GLUE suite. The experimental results are summarized in the newly added Fig. 5 on page 10 in the revised manuscript.
>
> Specifically, we consider a BERT-based classifier for the Stanford Sentiment Treebank (SST-2)
> dataset, which is a widely used benchmark for evaluating language-model fine-tuning in the field of natural language processing. The SST-2 dataset has positive and negative labels and contains a total of $67,349$ training samples. We partition the training
> dataset across five devices using a Dirichlet distribution with parameter $0.1$, creating a high degree of label-based heterogeneity among the devices. Our classifier employs a BERT-base encoder with a classification head. The BERT layers are frozen except for the last encoder layer, which is fine-tuned along with the classification head to adapt the representations to the SST-2 task. Both centralized gradient descent (GD) and our proposed Algorithm 1 are evaluated under this setup. Since the labels of the test dataset are not publicly available, we evaluate the test accuracy on the validation dataset instead. The results confirm the effectiveness of our algorithm. Please see the newly added section "Sentiment analysis on SST-2" on page 10 for details.
>
> Continued on Part IV

---

> ### Author Response · Authors · 2026-03-04
> **Response to Reviewer 4mwn---Part IV**
>
> Continued from Part III
>
> $\rm\color{orange}{@4mwn\ @Response\ to\ Weakness\ (W3)\ and\ Requested\ Changes\ (R3):}$ Thank you for the comment. In the revised manuscript, we have clarified that in the logistic regression experiment, each client $i$'s smoothness constant $L_{i}$ is obtained by calculating the largest eigenvalue of the local Hessian matrix using a standard numerical linear algebra routine, while in neural network training, each client $i$'s smoothness constant $L_{i}$ is obtained using a gradient difference approximation. Specifically,
>
> **1) Logistic regression experiment.** The local objective function is convex and twice differentiable, given by
>
> $f_i(x) = \frac{1}{m} \sum_{j=1}^m \log\left(1 + \exp(-b_j a_j^\top x)\right) + \frac{\mu}{2}||x||_2^2$,
>
> where $a_j \in \mathbb{R}^{300}$ denotes the feature vector of the $j$-th sample, $b_j \in${$-1,1$} is its corresponding label, and $m$ is the number of local data samples. In this case, the smoothness constant is characterized by the spectral norm of the local Hessian matrix, which is given by $\nabla^2 f_i(x) = \frac{1}{m} \sum_{j=1}^m \left[ \frac{\exp(b_j a_j^\top x)}{(1 + \exp(b_j a_j^\top x))^2} \right] a_j a_j^\top + \mu I$. Noting that $\frac{\exp(b_j a_j^\top x)}{(1 + \exp(b_j a_j^\top x))^2} \le \frac{1}{4}$, where the maximum value $1/4$ is attained when $a_j^\top x = 0$ (e.g., $x=0$), we obtain the bound $\nabla^2 f_i(x) \preceq \frac{1}{4m} \sum_{j=1}^m a_j a_j^\top + \mu I$, where $A \preceq B$ means that $ B-A$ is positive semidefinite. Therefore, for client $i$, the exact Lipschitz constant is given by the largest eigenvalue of $\left(\frac{1}{4m} \sum_{j=1}^m a_j a_j^\top + \mu I\right)$. This quantity can be computed efficiently using standard numerical linear algebra routines.
>
> **2) Neural network training.** Since a closed-form expression of the smoothness constant is generally unavailable in neural network training, we estimate $L_i$ numerically using a gradient difference approximation. Concretely, for each client $i$,  we fix a mini-batch of local data and randomly sample  model parameter vectors $x$ from a region that empirically covers typical parameter variations during training. For each sampled point, we construct a perturbed parameter vector $x+\varepsilon$ with a small perturbation $\varepsilon>0$, and compute the ratio:
>
> $\frac{||\nabla f_{i}(x+\varepsilon)-\nabla f_i(x)||}{||(x+\varepsilon)-x||}.$
>
> This procedure is repeated over tens of thousands of times with different random initializations, and the largest ratio observed across all runs is taken as an estimate of the local smoothness constant. This numerical approximation effectively estimates the local Lipschitz constant of the gradient and is commonly adopted when closed-form expressions are not available for deep neural networks.
>
> To clarify these points, in the revised manuscript, we have added Remark 1 on page 10, which is presented as follows:
>
> **"Remark 1. In logistic regression experiments, the smoothness constant $L_i$ for each client is computed directly from the local Hessian matrix using its largest eigenvalue. In neural network training, including CNN training and BERT fine-tuning, since a closed-form expression of $L_i$ is generally unavailable, we estimate it numerically using a gradient difference approximation. Specifically, a mini-batch of local data is fixed and multiple model parameters $x$ are randomly sampled from a region that empirically covers typical parameter variations during training. For each sampled parameter, a small perturbation $\varepsilon$ is applied and the ratio of gradient difference to parameter difference is computed. This procedure is repeated tens of thousands of times, and then the maximal value of the resulting ratios is taken as an estimate of the local smoothness constant $L_i$."**

---

### Review · Reviewer_8JfX · 2026-02-25

**Summary Of Contributions:**

This paper studies distributed optimization with a central server for minimizing a finite-sum of strongly convex and smooth functions. It questions the common assumption that centralized methods are inherently more efficient, showing instead that decentralized approaches can achieve faster convergence. The authors describe a distributed algorithm with a central server. They also formulate a Performance Estimation Problem (PEP) to characterize the worst-case performance of optimization algorithms. By solving the PEP as a semidefinite program (SDP), they obtain solutions with improved worst-case guarantees. Experimental results on the W8A, MNIST, and CIFAR-10 datasets are conducted to validate the theoretical findings.

Strengths: Applying PEP to analyze distributed optimization looks interesting.

Weakness: Beyond proposing an interesting statement, the paper does not appear to devote sufficient effort to a thorough theoretical investigation of the computational performance of distributed optimization to substantiate its main claim—that distributed methods are more efficient.

**Audience:**

Yes

**Audience Explanation:**

The statement seems interesting.

**Claims And Evidence:**

No

**Claims Explanation:**

The main claim of this paper is that “distributed methods (with a central server) can be more efficient than centralized methods,” which is quite a strong statement. Therefore, at least reasonable effort is required to make this claim convincing.

Here, I summarize the theoretical efforts made in this paper to support this statement.
1. The paper applies the PEP framework to a distributed algorithm with a central server, replacing the smoothness and strong convexity inequalities with corresponding conditions imposed on the local objective functions.
2. It solves the resulting SDP for the case of two clients and presents the numerical results in Figure 1.

I believe the theoretical efforts are rather limited and insufficient to support the paper’s main claim. There are additional analyses the authors could conduct to provide stronger justification. For example, given that the PEP formulation of Algorithm 1 is very similar to that of centralized gradient descent, could the authors directly compare the two and establish a formal relation between their respective optima?

Furthermore, the numerical SDP results are restricted to the case of two clients. It is unclear why this setting is representative of distributed optimization. Would not the regime with a much larger number of clients (N>>2) be more informative for comparing distributed and centralized algorithms?

**Requested Changes:**

Provide solid theoretical evaluations for comparison of the distributed and centralized algorithms.

---

> ### Author Response · Authors · 2026-03-04
> **Response to Reviewer 8JfX---Part I**
>
> $\rm\color{blue}{@8JfX\ @Response\ to\ weakness\ in\ thorough\ theoretical\ investigation:}$ We emphasize that our PEP-based results are *formally valid and mathematically rigorous*. Specifically,  they provide and compare the *exact* worst-case performance bounds of optimization algorithms over a *general class of functions* [R1]. This stands in contrast to conventional analyses, which typically rely on loose upper bounds that may be overly conservative and, at times, misleading [R2]. It is also fundamentally different from numerical simulations on *specific function instances*. In the revised version, we clarify this point by formally summarizing our main theoretical results in Theorem 1. In this theorem, we rigorously characterize the  performance of the proposed Algorithm 1 and compare it with centralized gradient descent. The result provides an explicit proof of the acceleration achieved by Algorithm 1, as illustrated in Figure 1(b). For the reviewer's convenience, we restate the theorem below.
>
> **"Theorem 1. (Convergence Acceleration of Algorithm 1)    Consider the optimization problem defined in (1), where the function $f$ belongs to the class $\mathcal{F}_{\mu, L}$, characterized by the smoothness parameter $L$  and strong convexity parameter $\mu$. Let $\cal{A}_1$ denote Algorithm 1 with local step sizes $\frac{1}{L_i}$, and $\cal{A}_c$ denote centralized gradient descent with a uniform step size $\frac{1}{L}$. For the performance metric**
>
> $P\_{\\mathcal{A}}=\\max\_{f \\in \\mathcal{F}\_{\\mu,L}}\\{\\frac{1}{N}\\sum\_{i=1}^{N}||x\_{i,\\mathcal{A}}^{k} - x^{\\ast}||^2\\}$,
>
> **which measures the worst-case optimization error at iteration $k$ over all $f\in\mathcal{F}_{\mu, L}$, we consider the ratio**  $P\_{\\mathcal{A}\_1}$/$P\_{\\mathcal{A}\_c}$ **as the relative acceleration of Algorithm 1 compared with centralized gradient descent (with values below $1$ indicating acceleration and smaller ratios indicating stronger acceleration). Figure 1(b) gives the exact ratio values under different function classes, and the ratio values being strictly less than $1$ for all tested function classes $\mathcal{F}_{\mu, L}$ in the heterogeneous setting confirm that Algorithm 1 with local step sizes achieves a strictly faster convergence rate than centralized gradient descent under all considered function classes."**
>
> ***"Proof.*** **First, we have $P_{\mathcal{A}} < \infty$ for any finite $k$, which implies that the worst-case optimization error of Algorithm 1 over the function class $\mathcal{F}_{\mu, L}$ is always finite for any $k$.**
>
> **Therefore, the set** $\\arg\\max\_{f\\in\\mathcal{F}\_{\\mu, L}}\\{\\frac{1}{N}\\sum\_{i=1}^{N}||x\_{i,\\mathcal{A}}^{k} - x^{\\ast}||^2\\}$ **is guaranteed to be nonempty. This means that there exists at least one function instance $f_{\mathcal{A}}^*$ such that,  Algorithm 1, when executed with step-size scheme $\mathcal{A}$, indeed attains its worst-case performance at iteration $k$.**
>
> **For any such maximizer**   $f\_{\\mathcal{A}}^{\ast}\\in\\arg\\max\_{f\\in\\mathcal{F}\_{\\mu, L}}\\{\\frac{1}{N}\\sum\_{i=1}^{N}||x\_{i,\\mathcal{A}}^{k} - x^{\ast}||^2\\}$, **the value** $P_{\cal{A}}$ **returned by the PEP is therefore an *exact*, not merely upper-bounding, characterization of the true worst-case error over** $\mathcal{F}_{\mu, L}$. **In other words, the PEP does not produce a loose bound: it identifies a specific problem instance on which Algorithm 1 performs exactly as poorly as the bound indicates.**
>
> **With this interpretation, the ratio** $\\frac{P\_{\\mathcal{A}\_1}}{P\_{\\mathcal{A}\_c}}$ **directly compares the exact worst-case optimization errors of Algorithm 1 versus centralized gradient descent.** **A value** $\\frac{P\_{\\mathcal{A}\_1}}{P\_{\\mathcal{A}\_c}}<1$ **therefore means that—in the worst case—Algorithm 1 has strictly smaller optimization error than centralized gradient descent at iteration $k$. Hence, the ratio values in Figure 1(b) quantitatively capture the exact speed gains achieved by Algorithm 1."**
>
> [R1] Adrien B Taylor, Julien M Hendrickx, and François Glineur. Exact worst-case performance of first-order methods for composite convex optimization. *SIAM Journal on Optimization*, 27(3):1283–1313, 2017.
>
> [R2] Erwan Meunier and Julien M. Hendrickx. Several performance bounds on decentralized online optimization are highly conservative and potentially misleading. *IEEE 64th Conference on Decision and Control (CDC)*, IEEE, pp. 6201-6207, 2025.
>
> Continued on Part II

---

> ### Author Response · Authors · 2026-03-04
> **Response to Reviewer 8JfX---Part II**
>
> Continued from Part I
>
>
> $\rm\color{blue}{@8JfX\ @Response\ to\ the\ comment\ about\ comparing\ Algorithm\ 1\ to\ centralized\ gradient\ descent:}$ Thank you for your  suggestion. However, we would like to argue that we indeed have the suggested comparison between Algorithm 1 and centralized gradient descent. Specifically, in Figure 1(a), we provide a PEP based theoretical ablation study of our proposed method, demonstrating that even when the switching time is not carefully tuned, our algorithm still achieves stronger performance than centralized gradient descent. Moreover, Figure 1(b) shows that our results hold across a wide range of values for $L$. As illustrated in Figure 1(c), the theoretical guarantees are not restricted to the two-agent case and can be extended to larger networks  with more agents.
>
> Again, we emphasize that our PEP-based results are *formally valid and mathematically rigorous*. In particular, they establish and compare the exact worst-case performance bounds of optimization algorithms over a *general class of functions* [R1]. This is fundamentally different from numerical simulations on *specific function instances*.
> Using PEP, we theoretically and rigorously compare Algorithm 1 with centralized gradient descent. In Figure 1(a), a clear *exact* worst-case bound improvement of Algorithm 1 over centralized gradient descent can be seen. In addition, Theorem 1 in the revised manuscript formally and rigorously characterizes the convergence speed gain of Algorithm 1 over centralized gradient descent. The corresponding results are presented in Figure 1(b), where each value less than 1 indicates a speedup of Algorithm 1 compared with centralized gradient descent.
>
> $\rm\color{blue}{@8JfX\ @Response\ to\ the\ comment\ about\ the\ case\ of\ two\ clients:}$ Thank you for the comment.
> In addition to the results for two clients shown in Figure 1(a) and Figure 1(b), we also demonstrate the validity of our findings for up to 16 agents in Figure 1(c). Although the high computational complexity of SDP makes it challenging to obtain a PEP-based rigorous performance characterization for even larger-scale networks, our empirical results consistently support the theoretical claims in this setting (see the newly added experimental evaluation on varying network sizes using the MNIST dataset in Fig. 7).
>
> Continued on Part III

---

> ### Author Response · Authors · 2026-03-04
> **Response to Reviewer 8JfX---Part III**
>
> Continued from Part II
>
> $\rm\color{blue}{@8JfX\ @Response\ to\ Requested\ Changes\ to\ provide\ theoretical\ comparison\ with\ centralized\ gradient\ descent :}$ We emphasize that the PEP-based results are *formally valid and mathematically rigorous*, as they provide and compare the *exact* worst-case bounds for optimization algorithms over a *general class of functions* [R1]. To make the theoretical conclusions of the paper clear, in the revised version, we add Theorem 1 to formally compare our decentralization based distributed algorithm with centralized gradient descent using PEP. For the reviewers' convenience, we restate Theorem 1 as follows:
>
> **"Theorem 1. (Convergence Acceleration of Algorithm 1)    Consider the optimization problem defined in (1), where the  function $f$ belongs to the class $\mathcal{F}_{\mu, L}$, characterized by the smoothness parameter $L$  and strong convexity parameter $\mu$. Let $\cal{A}_1$ denote Algorithm 1 with local step sizes $\frac{1}{L_i}$, and $\cal{A}_c$ denote centralized gradient descent with a uniform step size $\frac{1}{L}$. For the performance metric**
>
> $P\_{\\mathcal{A}}=\\max\_{f \\in \\mathcal{F}\_{\\mu,L}}\\frac{1}{N}\\sum\_{i=1}^{N}||x\_{i,\\mathcal{A}}^{k} - x^*||^2$,
>
> **which measures the worst-case optimization error at iteration $k$ over all $f\in\mathcal{F}_{\mu, L}$, we consider the ratio**  $P\_{\\mathcal{A}\_1}$/$P\_{\\mathcal{A}\_c}$ **as the relative acceleration of Algorithm 1 compared with centralized gradient descent (with values below $1$ indicating acceleration and smaller ratios indicating stronger acceleration). Figure 1(b) gives the exact ratio values under different function classes, and the ratio values being strictly less than $1$ for all tested function classes $\mathcal{F}_{\mu, L}$ in the heterogeneous setting confirm that Algorithm 1 with local step sizes achieves a strictly faster convergence rate than centralized gradient descent under all considered function classes."**
>
> ***"Proof.*** **First, we have $P_{\mathcal{A}} < \infty$ for any finite $k$, which implies that the worst-case optimization error of Algorithm 1 over the function class $\mathcal{F}_{\mu, L}$ is always finite for any $k$.**
>
> **Therefore, the set** $\\arg\\max\_{f\\in\\mathcal{F}\_{\\mu, L}}\\{\\frac{1}{N}\\sum\_{i=1}^{N}||x\_{i,\mathcal{A}}^{k} - x^{\\ast}||^2\\}$ **is guaranteed to be nonempty. This means that there exists at least one function instance $f_{\mathcal{A}}^*$ such that,  Algorithm 1, when executed with step-size scheme $\mathcal{A}$, indeed attains its worst-case performance at iteration $k$.**
>
> **For any such maximizer**   $f\_{\\mathcal{A}}^{\\ast}\\in\\arg\\max\_{f\\in\\mathcal{F}\_{\\mu, L}}\\{\\frac{1}{N}\\sum\_{i=1}^{N}||x\_{i,\\mathcal{A}}^{k} - x^{\\ast}||^2\\}$, **the value** $P_{\cal{A}}$ **returned by the PEP is therefore an *exact*, not merely upper-bounding, characterization of the true worst-case error over** $\mathcal{F}_{\mu, L}$. **In other words, the PEP does not produce a loose bound: it identifies a specific problem instance on which Algorithm 1 performs exactly as poorly as the bound indicates.**
>
> **With this interpretation, the ratio** $\\frac{P\_{\mathcal{A}\_1}}{P\_{\\mathcal{A}\_c}}$ **directly compares the exact worst-case optimization errors of Algorithm 1 versus centralized gradient descent.** **A value** $\\frac{P\_{\\mathcal{A}\_1}}{P\_{\\mathcal{A}\_c}}<1$ **therefore means that—in the worst case—Algorithm 1 has strictly smaller optimization error than centralized gradient descent at iteration $k$. Hence, the ratio values in Figure 1(b) quantitatively capture the exact speed gains achieved by Algorithm 1."**
>
> [R1] Adrien B Taylor, Julien M Hendrickx, and François Glineur. Exact worst-case performance of first-order methods for composite convex optimization. *SIAM Journal on Optimization*, 27(3):1283–1313, 2017.

---

> > ### Comment · Reviewer_8JfX · 2026-04-09
> >
> > The authors aim to compare the theoretical convergence guarantees of a distributed optimization method and a centralized optimization method using PEP. However, the comparisons appear to rely entirely on numerical solutions obtained by reformulating the PEP as an SDP, without accompanying human-readable analytical insights. As a result, I find it difficult to interpret Theorem 1 as a “theoretical result”, since it largely reiterates the PEP formulation and draws conclusions about the relative efficiency of distributed methods based primarily on numerical evidence.
> >
> > Given my background, I regret that I am not well positioned to thoroughly evaluate this work. To the best of my knowledge, PEP is often used as a supporting tool to derive interpretable, human-readable proofs, which I personally find more accessible. Since this paper seems to rely primarily on (potentially selective) numerical results, I find it challenging to assess the validity and generality of its conclusions.

---

> ### Author Response · Authors · 2026-04-09
> **Response to Official Comment by Reviewer 8JfX**
>
> We appreciate this additional comment. However, we respectfully disagree with the comment that the paper "rely primarily on (potentially selective) numerical results."  We emphasize that Performance Estimation Problems (PEP) constitute a **rigorous** worst-case analysis framework, rather than merely a numerical experiment. Although the resulting bounds are obtained by solving a semidefinite program (SDP), these solutions correspond to **certified worst-case guarantees over an infinite-dimensional class of problems**. In this sense, the conclusions drawn from PEP are **theoretical and hold universally, rather than being empirical observations tied to specific instances**.
>
> We include an example to clarify how our approach differs from standard numerical simulations. Numerical experiments typically evaluate performance on a specific function instance, for example $ f (x) = x^2$. In contrast, the PEP-based comparison is fundamentally different. Instead of fixing a function instance like $ f (x) = x^2$, it considers an entire function class, such as \\(\\mathcal{F}\_{\\mu, L}\\), which contains **infinitely many functions** that are $\\mu$-strongly convex and $L$-smooth. The PEP framework then provides the **exact worst-case optimization error over this entire function class**, rather than performance on a particular function instance.
>
> Regarding Theorem 1, our intention is to formalize the comparison between distributed and centralized methods within the PEP framework. Although the theorem is instantiated via an SDP, it encodes exact worst-case guarantees over **a class of functions and therefore constitutes a valid theoretical** result. This paradigm is now well established in optimization theory and has been successfully used to derive tight bounds that are often difficult or impossible to obtain in analytical form (see [R3, R4, R5] below for representative examples).
>
> We acknowledge, however, that the current presentation may give the impression that the conclusions are numerical. To address this concern, we will revise the paper to clarify that the SDP solutions correspond to provable worst-case guarantees over **an infinite-dimensional class of problems**, rather than being empirical observations tied to specific instances.
>
> Also, we agree that in the early development of the PEP methodology, PEP was often used as a supporting tool to derive analytical proofs [R1, R2]. However, our work belongs to a more recent and increasingly popular line of research [R3, R4, R5], where PEP is used to gain insights into algorithm design and parameter selection---insights that are not accessible through traditional analytical, human-readable proofs. While it would be desirable to establish our results using classical analytical techniques, we rely on the PEP framework precisely because it enables the derivation of exact performance bounds that cannot be obtained through traditional analysis.
>
> [R1] A. B. Taylor, J. M. Hendrickx, and F. Glineur, “Exact worst-case performance of first-order methods for composite convex optimization,” SIAM J. Optim., vol. 27, no. 3, pp. 1283–1313, 2017.
>
> [R2] E. de Klerk, F. Glineur, and A. B. Taylor, “On the worst-case complexity of the gradient method with exact line search for smooth strongly convex functions,” Optim. Lett., vol. 11, pp. 1185–1199, 2017.
>
>
> [R3] Erwan Meunier and Julien M. Hendrickx. Several performance bounds on decentralized online optimization are highly conservative and potentially misleading. IEEE 64th Conference on Decision and Control (CDC), IEEE, pp. 6201-6207, 2025.
>
> [R4] S. Colla and J. M. Hendrickx, “Automatic performance estimation for decentralized optimization,” IEEE Trans. Autom. Control, vol. 68, no. 12, pp. 7136–7150, 2023.
>
> [R5] S. D. Gupta, B. P. G. Van Parys, and E. K. Ryu, “Branch-and-bound performance estimation programming: A unified methodology for constructing optimal optimization methods,” Math. Program., vol. 204, pp. 567–639, 2024.

---

### Decision · Action_Editor_pesj · 2026-04-27

**Recommendation:** Reject

**Additional Comments:**

For a resubmission, the authors should: (1) compare against stronger centralized baselines including SGD with momentum and Adam, (2) provide analytical insight into when and why local step-size adaptation yields acceleration, (3) analyze robustness to $L_i$ estimation errors theoretically, and (4) consider reframing the contribution as local geometric adaptation rather than decentralization.

**Audience:**

Yes

**Audience Explanation:**

The paper uses per-client step sizes tuned to local smoothness constants in a centrally coordinated or server-assisted federated setup, with a switch to uniform step sizes near convergence to eliminate bias, and validates the acceleration over vanilla centralized GD via PEP-based worst-case analysis. The paper is interesting to researchers in the field of Federated Learning.

**Claims And Evidence:**

No

**Claims Explanation:**

The paper still contains some major weaknesses as indicated below and needs substantial revision to be accepted at TMLR.
1. The paper's primary comparison is against vanilla full-batch centralized GD with a uniform step size, a method rarely used in practice. While the authors added comparisons with FedOpt variants (FedAdam, FedYogi, FedAdagrad, FedAvgM) in the revision, these  methods are not specifically designed for heterogeneous data unlike FedProx or SCAFFOLD which would be more relevant baselines given the paper's focus on data heterogeneity. Moreover, they still do not compare against standard centralized optimizers like SGD with momentum or Adam, which are the actual workhorses of modern ML training. The claimed acceleration may diminish or vanish against these stronger baselines.
2. As pointed out by 2/3 reviewers, the paper lacks analytical insight into why heterogeneous smoothness constants lead to acceleration. For a paper claiming to overturn a longstanding assumption, a deeper theoretical understanding is needed to make the claim convincing.
3.  The algorithm requires per-client smoothness constants $L_i$, but for neural networks these are estimated via a heuristic gradient-difference procedure done once before training. While the authors show empirically that different estimates yield similar results, there is no theoretical analysis of how errors in $L_i$ estimation affect convergence, as also noted by Reviewer 4mwn.
4. The framing of "decentralization accelerates optimization" is misleading. The algorithm is server-assisted (federated), and the acceleration comes from exploiting per-partition smoothness via local step sizes, a form of preconditioning that could equally be applied in a centralized setting by intentionally partitioning data, as the authors themselves acknowledge in Section 3.1. The core contribution is local step-size adaptation rather than decentralization, in my opinion.

**Resubmission Of Major Revision:**

The authors may consider submitting a major revision at a later time.